# OWMM-Agent: Open World Mobile Manipulation With Multi-modal Agentic Data Synthesis

**Junting Chen**[12*]   **Haotian Liang**[13*]   **Lingxiao Du**[1]   **Weiyun Wang**[1]   **Mengkang Hu**[14]
**Yao Mu**[15]   **Wenhai Wang**[1]   **Jifeng Dai**[16]   **Ping Luo**[14]   **Wenqi Shao**[1†]   **Lin Shao**[2]
[1]Shanghai AI Laboratory    [2]School of Computing, National University of Singapore    [3]USTC
[4]The Univeristy of Hongkong    [5]Shanghai Jiaotong University    [6] Tsinghua University

## Abstract

The rapid progress of navigation, manipulation, and vision models has made mobile manipulators capable in many specialized tasks. However, the open-world mobile manipulation (OWMM) task remains a challenge due to the need for generalization to open-ended instructions and environments, as well as the systematic complexity to integrate high-level decision making with low-level robot control based on both global scene understanding and current agent state. To address this complexity, we propose a novel multi-modal agent architecture that maintains multi-view scene frames and agent states for decision-making and controls the robot by function calling. A second challenge is the hallucination from domain shift. To enhance the agent performance, we further introduce an agentic data synthesis pipeline for the OWMM task to adapt the VLM model to our task domain with instruction fine-tuning. We highlight our fine-tuned OWMM-VLM as the first dedicated foundation model for mobile manipulators with global scene understanding, robot state tracking, and multi-modal action generation in a unified model. Through experiments, we demonstrate that our model achieves SOTA performance compared to other foundation models including GPT-4o and strong zero-shot generalization in real world. The project page is at `https://hhyhrhy.github.io/owmm-agent-project`.

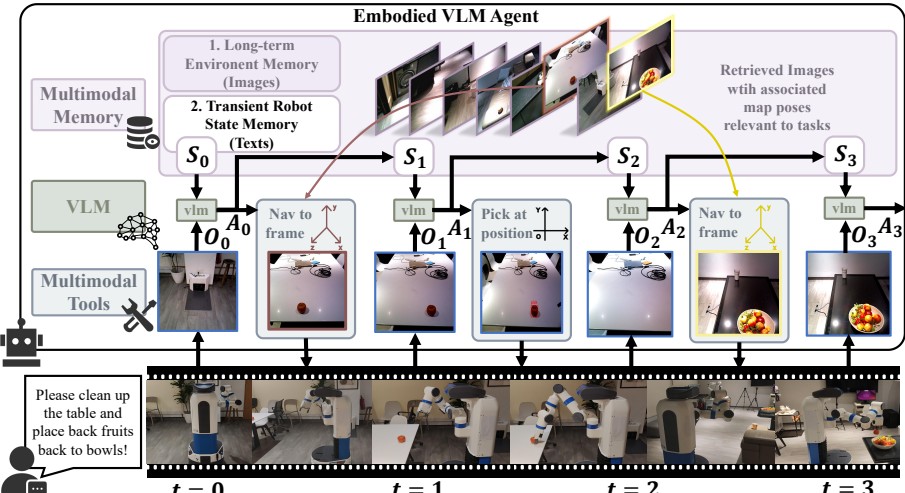

Figure 1: **OWMM-Agent Operates Fetch Robot for Tidying Task.** *OWMM-Agent* receives natural language instructions and leverages both long-term environment memory (scene images) and transient robot state memory (textual summary) to generate sequential multi-modal actions to finish the task. By **multi-turn, multi-image, and multi-modal** VLM reasoning, the agent conducts global scene aware reasoning, updates robot state memory, and actuates itself to desired coordinates without any other learning-based models in unstructured environments.

---

*Equal Contributions

†Corresponding Author

39th Conference on Neural Information Processing Systems (NeurIPS 2025).

# 1 Introduction

The vision of generalist home assistant robots has brought open-world mobile manipulation (OWMM) to the forefront of embodied AI research [41, 37, 40, 43, 28]. OWMM tasks require mobile manipulators to interpret open-ended natural language instructions and operate in unstructured, previously unseen environments. Although advancements in navigation, manipulation, and vision models have effectively enabled mobile manipulators to perform many specialized tasks under constraints, achieving robust autonomy in these settings remains challenging.

A central difficulty in OWMM is the need for comprehensive global scene understanding and reasoning conditioned on natural language instructions and agent state. On one hand, prior approaches often construct 2D semantic maps [30] or 3D semantic fields with CLIP-based features [19, 28], retrieving targets by computing embedding distances between the semantic map and language instructions. While these methods have enabled progress, they are limited by the capacity of embedding models, which can struggle with complex, compositional instructions, compared to foundational generative models like large language models (LLM) or vision-language models (VLM). Additionally, they often require time-consuming dense 3D reconstruction, making them less suitable for complex, open-ended and dynamic environments. On the other hand, the recent advances in LLMs and VLMs, with strong generalization capability, versatility, and reasoning capability, offer promising opportunities and potentially a fundamental pathway to solve all sorts of scene understanding, task planning, and robot control issues in open-world intelligent robot systems [20, 15].

Based on the aforementioned observations, we propose a novel VLM agent framework, *OWMM-Agent*, to address these challenges and leverage the power of VLMs for OWMM task. More specifically, we formulate the high-level OWMM task for the internal VLM model as a multi-turn, multi-image, and multi-modal reasoning problem. The VLM model generates end-to-end chain-of-thought (CoT) thinking process, tracked agent states, and multi-modal actions with coordinates based on all raw multi-modal input. Then the agent calls the coordinate-based planners to actuate the robot. Our approach is built on two insights: 1) We do not need the detailed geometric representation of the environment for instruction-conditioned global scene understanding, and we could easily access precise and even dynamic geometric information when the robot moves to the task-relevant local region. 2) By leveraging the strong vision-language grounding capabilities, we can effectively bridge the high-level reasoning process in language and low-level robot control targets in coordinates, with the help of 2D-to-3D reverse projection.

However, directly applying pre-trained VLMs to our embodied agent presents challenges of domain shift: 1) **Rare grounding tasks**: Robotic planners and controllers require multi-modal inputs, including both tools and coordinates in the visual space for robot control. The base models could be powerful for object-centric grounding such as detecting novel objects, but they suffer in other grounding tasks including detecting non-blocked navigable areas in the ego-centric image. 2) **State tracking**: The agent must infer and track its own state from observations and history records to make contextually appropriate decisions. 3) **Embodiment priors**: Effective decision-making in egocentric settings demands strong embodiment-dependent priors, such as knowledge of the robot's kinematic constraints, such as maximum reach for picking actions.

To address the problem of domain adaptation, we further introduce an agentic data synthesis pipeline tailored for OWMM, to generate large-scale and instruction-driven episodes that teach the VLM agent to track its state, reason over multi-view observations, and generate multi-modal action affordances grounded in both the global scene and the agent's embodiment. This pipeline minimizes human annotation effort by utilizing predefined task sequence templates and ground-truth symbolic world representations from simulation. With extensive experiments in simulation, we demonstrate that OWMM-VLM consistently outperforms baseline models. In the real-world experiment, we find that our model has strong zero-shot generalization to real-world observations, with $27/30 = 90\%$ action generation success rate on our fetch robot in the lab environment, even being fine-tuned on the simulated data. We also provide ablation studies on models and dataset analysis to provide insights into the model design and training data construction. In summary, our contributions are as follows:

- We propose *OWMM-Agent*, a unified VLM-based agent architecture for open-world mobile manipulation, capable of global scene understanding, state tracking, and end-to-end action generation.

- We introduce a simulation-based agentic data synthesis pipeline that enables scalable data collection for instruction fine-tuning for domain adaptation with minimized human effort, with detailed analysis on the quality of the generated dataset.

- We introduce a foundation model for OWMM, capable of multi-image reasoning and executable multi-modal action generation, with extensive experiments analyzing the model's performance.

## 2 Related Works

**Open World Mobile Manipulation**

Open-vocabulary Mobile Manipulation (OVMM) focuses on navigating and manipulating novel objects in unseen environments with language instructions. Referred to as *Open Vocabulary Mobile Manipulation* (OVMM) by [40, 43, 19] or *Open World Mobile Manipulation* (OWMM) by [29, 41, 37], we use the term OWMM for this paper. In our formulation, "open-world" refers to semantic diversity—the ability to generalize to unseen scenes, novel object categories, and diverse instances—rather than unconstrained physical exploration without any prior environmental information. This terminology aligns with the HOMERobot challenge [40].

The original OWMM baseline and Melnik et al. [23] assume the agent starts without scene observation and must explore to build a representation for decision-making. Recent works [19, 28, 43] suggest a two-stage approach: first using SLAM [7] to create 3D semantic maps, then performing OVMM using open-vocabulary models like GPT-4V and GPT-4o [11].

Zhi et al. [43] introduces *COME-robot*, a closed-loop OVMM framework using GPT-4V for reasoning and replanning, producing code for preset functions and object captions as in *Code-as-Policy* [17]. Unlike relying on pre-trained skill models requiring inputs like skill names and object captions, our model directly produces target positions for position-based motion planners and controllers.

**Vision-Language Navigation and Interactive Embodied Tasks**

Our work is related to the broader vision-language navigation (VLN) literature, which has explored goal-oriented navigation and manipulation in indoor environments. REVERIE [27] introduced remote embodied visual referring expressions, requiring agents to navigate and ground target objects based on natural language descriptions. SOON [44] proposed scenario-oriented object navigation with graph-based exploration strategies. These works primarily focus on navigation with discrete action spaces and viewpoint selection. ALFRED [31] presented a benchmark for interpreting grounded instructions for everyday tasks, while FILM [24] proposed modular methods for following instructions in language. Recent work has also explored topological planning with transformers for VLN [4]. Our OWMM setting differs from these VLN works in key aspects: (1) **Continuous action space**: We use continuous positional control requiring precise affordance grounding rather than discrete viewpoint selection. (2) **Direct object interaction**: Our agent physically manipulates objects with low-level joint control, whereas VLN tasks use simplified interactions or only predict bounding boxes. (3) **Sim-to-real transfer**: Our framework is validated on real robotic hardware. (4) **Unified reasoning**: We formulate the task as multi-modal, multi-turn reasoning within a single VLM, integrating perception, planning, and action generation.

**Large Foundational Models for Robotics**

Recent advances in large fundamental models show significant potential in robotic control and generalization. One major research focus is to adapt pre-trained Visual Language Model (VLM) to robot scenario. RoboPoint [42] introduces a synthetic data pipeline for instruction-tuning VLMs in robotics, supporting accurate spatial affordance prediction in object manipulation and navigation. MOKA [18] uses a novel VLM approach in robotic manipulation with point-based affordance and motion representation, using visual prompts to turn key points and waypoint predictions into visual question-answering tasks for VLMs. Our proposed model *OWMM-VLM* also falls into this category.

The other popular research topic is Vision-Language-Action (VLA) models, focusing on using relatively smaller transformer backbones to directly generate robotic actions with high frequency. OpenVLA [14] is a 7B-parameter open-source model trained on 970,000 real-world demonstrations using Llama 2 [34] architecture, excelling in general manipulation tasks. Octo [33] advances generalist robot policies, handling language commands and goal images while adapting quickly to new inputs and actions with standard GPUs. $\pi_0$ [1] presents a flow-matching architecture based on a

pre-trained VLM, excelling in dexterous tasks. These models mark significant progress in making robotic systems more versatile, scalable, and adaptable to different trajectories.

# 3 Methodology

In this section, we introduce the definition of OWMM in section 3.1. After that, we elaborate on the agent framework in section 3.2. Finally, the training method of OWMM-VLM is presented in section 3.3. The overview of our method is shown in Figure 2.

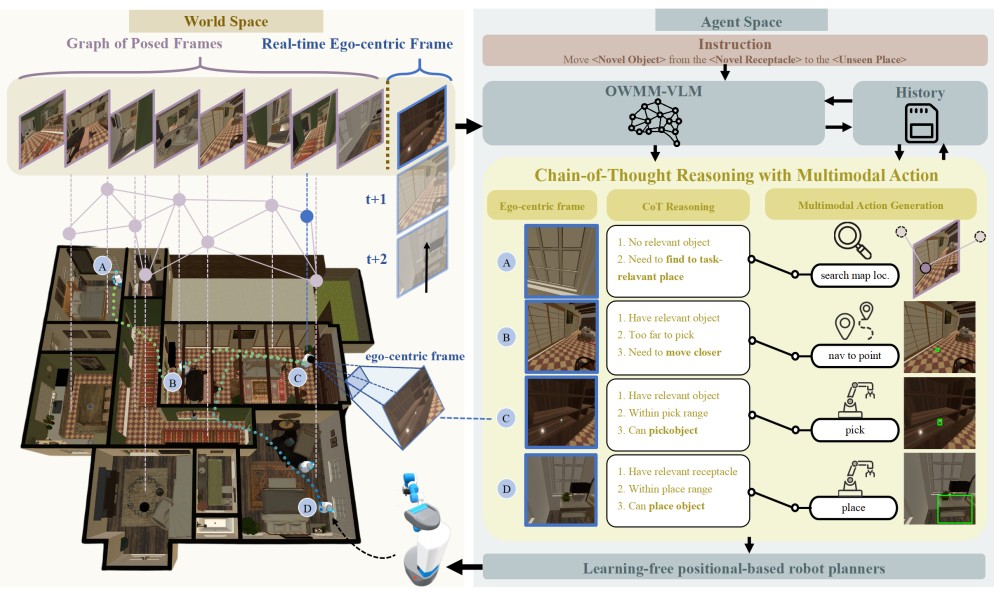

Figure 2: **The Overview of OWMM Agent Framework.** The left panel represents the world space, including a graph of posed frames generated during the pre-mapping phase and a real-time egocentric frame captured by the robot. The right panel showcases the Agent Space, where OWMM-VLM processes task instructions, robot history, and visual inputs to perform chain-of-thought reasoning and generate high-level actions with region coordinates, which are then sent to robot planners for navigation and manipulation.

## 3.1 OWMM Task Definition

Following the common OVMM/OWMM problem setting [40, 19, 37], the robot needs to follow the instruction in the pattern of "Move ⟨A⟩(in ⟨B⟩) and place it on/in ⟨C⟩", where ⟨A⟩⟨B⟩⟨C⟩ are novel objects/initial receptacles/goal receptacles in the unseen environment from the training data.

Following the problem setting in [19, 28], we assume a pre-mapping phase separating active exploration and the SLAM module from the OWMM task focus. This is practical, as most robotic vacuums automate room mapping before cleaning. The pre-mapping implementation details are provided in Appendix C.1.

Thus, we introduce a pose graph $G$ and associated RGB images $\mathcal{I}$ as the output of the pre-mapping stage on the basis of [40], and define our OWMM problem as follows: In an OWMM task episode of max timestep $T$, at each timestep $t, 0 \leq t \leq T$, an agent takes inputs composed of 1) a natural language instruction $\mathcal{L}$; 2) a pre-mapping camera pose graph $G = \{V, E\}$ of $n$ poses, where $V = v_0, \ldots, v_n$ edges are not used;

3) and associated RGB images $\mathcal{I} = \{I_0, \ldots, I_n\}$, each image $I_i \in \mathbb{R}^{3 \times w \times h}$ are taken at head camera view pose $v_i$ in $G$;

4) the agent's current head camera RGB image $I_t^c$ and depth image $D_t^c$.

With these inputs, an agent needs to generate a low-level continuous action $a_t$ that directly actuates the robot kinematically, including joint velocities of the robot arm and the base velocity of the robot. Let's $F_{agent}$ note the logical function of the agent policy model, and we have

$$\mathbf{a}_t = F_{agent}(\mathcal{L}, G, \mathcal{I}, I_t^c, D_t^c, \mathbf{x}_t), \tag{1}$$

where $\mathbf{x}_t$ stands for the robot state at time $t$.

### 3.2 OWMM Agent

Running large VLM models at 25Hz and gathering sufficient data for training a generalist VLA model from open-set language and visual observations remain challenging. To address this latency issue in the OWMM agent, we have the large VLM to produce high-level actions. The agent employs a unified VLM model $F_{vlm}$ to convert visual and lingual inputs into action types and positional commands, using a classical planner for navigation and a motion planner for manipulation, similar to Rekep[10]. The model's output represents a high-level action $A_t$ spanning several simulation steps, while planners resolve trajectories and low-level actions $\mathbf{a}_t$ for each step.

$$A_t, \mathcal{H}_t = F_{vlm}(\mathcal{L}, G, \mathcal{I}, I_t^c, \mathcal{H}_{t-1}), \tag{2}$$

$$\mathbf{a}_t = A_t(\mathbf{x}_t, D_t^c), \tag{3}$$

where $\mathcal{H}_t, \mathcal{H}_{t-1}$ are the high-level robot history, updated by the VLM model itself. $a_t = A_t(\mathbf{x}_t, D_t^c)$ indicates that the high-level action itself can be converted to executable code with the action handle linked to different planners and positional targets. In this regard, part of the high-level action $A_t$ can be seen as a special type of language model program, as proposed in [17]. Then the linked planner takes the state of the robot $\mathbf{x}_t$, and point clouds converted from depth map $D_t^c$ as an additional input to calculate the low-level action $a_t$.

To translate high-level action $A_t$ into low-level action $a_t$, the agent has a path planner [8] for navigation and a motion planner [32] for arm manipulation. These planners generate waypoints that satisfy mechanical constraints for base chassis and arm joints through sampling-based methods. There is also a gripper controller to grasp/ungrasp the object. The high-level actions that aim to actuate the robot will be associated with planners and controllers through predefined functions.

### 3.3 OWMM-VLM model

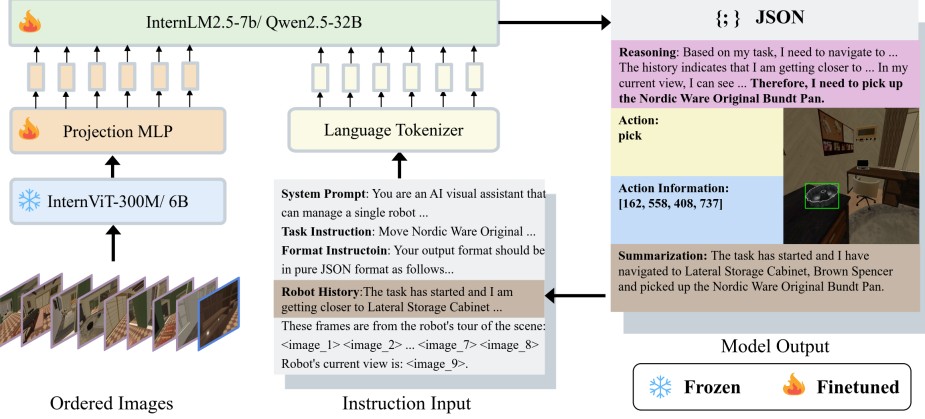

Figure 3: **Overview of OWMM-VLM.** Our model is fine-tuned on InternVL-2.5[5], comprising a ViT, a 2-layer projection MLP, and a LLM. During training, ViT parameters are frozen while the projection MLP and the LLM parameters are trainable. The model is required to generate multi-modal actions in JSON format conditioned on scene images, task instructions, and robot history.

Intuitively, a VLM model requires three core multi-modal capabilities to accomplish the OWMM task:

(1) **Image Retrieval.** Given the graph of posed frames and an egocentric frame, the VLM model needs to retrieve a posed frame that contains the relative objects or receptacles that the robot needs to navigate to.

(2) **Ego-centric Decision-making.** Given multiple posed frames and an egocentric frame, the VLM model needs to decide which action to conduct based on the task context, robot history, and current egocentric observation. This capability is closely associated with the idea of spatial intelligence[39], that VLM models should understand the spatial relationship between themselves and the scene objects in order to make decisions on actions.

(3) **Affordance Grounding.** If the agent decides to interact with the near surroundings perceived in the egocentric frame, it should also generate the target positions that correspond to the intention of the task.

Following this insight, we train a versatile VLM model that takes the task instruction $\mathcal{L}$, multimodal observations $\mathcal{I}, I_t^c$, and history $\mathcal{H}_{t-1}$, and generates all high-level actions. We design four types of high-level actions: 1) Posed image retrieval, 2) Navigate to point, 3) Pick, and 4) Place, which are associated with planners and the grip controller. However, due to the extended time horizon of the OWMM task, simply generating the executable action is insufficient. We instruct the VLM model to monitor the state through robot history and to infer the subsequent action by considering both the history and the present observations. Figure 3 demonstrates our model architecture as well as its input and output.

### 3.3.1 Chain-of-Thought Reasoning Design

To address the visual hallucination challenges faced by pre-trained VLMs in OWMM tasks, we design a structured Chain-of-Thought (CoT) reasoning framework. During our baseline evaluation, we observed three key hallucination-related failure modes: (i) **Error location outputs**: Base models achieve very low affordance success rates (0.05-0.18) due to incorrect object detection or mismatched bounding-box outputs; (ii) **Multi-image hallucination**: Models frequently hallucinate on images where target objects are absent, achieving only 1.27% accuracy on 8-image retrieval; (iii) **Insufficient long-horizon reasoning**: Models fall into dead loops due to limited ability to track historical context and robot state.

Our CoT reasoning approach generates structured reasoning chains that include: 1) *Task instruction reasoning and summarization* for decision-making; 2) *Perception and grounding* of current egocentric-view and scene images, integrating visual information to support task decisions (e.g., determining whether the agent is close enough to an object for interaction); 3) *Task decision output* with execution targets/coordinates as bounding boxes; 4) *Summarization* of decisions and actions for next-step input.

This design allows OWMM-VLM to acquire patterns for task comprehension, scene perception, and decision-making from structured training data. Crucially, the model summarizes historical context after each decision, enabling each subsequent step to jointly reason over prior history and current observations. This structured reasoning, learned through supervised fine-tuning on CoT-annotated data, enables the model to track task progress and avoid repetitive actions or dead loops. As shown in Appendix G, removing reasoning and summarization capabilities leads to substantial performance degradation across all metrics.

For more details on model implementation, see Appendix C.

## 4 Dataset

In this section, we elaborate on our data construction pipeline and quality verification method in section 4.1. A detailed analysis of the data is provided in section 4.2.

### 4.1 Agentic Data Synthesis Pipeline

Effective OWMM-VLM model training requires comprehensive ground-truth annotations covering navigation, object grasping, and manipulation affordances with full contextual understanding. Previous research [42, 21] often generates question-answer pairs from images or videos, but lacks comprehensive action sequence representations and necessary affordance information for multi-step reasoning.

To address this challenge, we developed an automated agentic data synthesis pipeline that generates both action sequences and structured chain-of-thought annotations. Our pipeline consists of four key stages:

Table 1: **Dataset Overview for Instruction Fine-tuning.** Our dataset consists of four subsets, each corresponding to one of the four primary task actions: *Pick*, *Place*, *Navigate to Point*, and *Search Scene Frame*. The dataset is designed to encompass diverse scenarios and objects, ensuring comprehensive coverage of open-world mobile manipulation tasks.

| Task Action | Pick | Place | Nav to point | Search scene frame |
|---|---|---|---|---|
| |  |  |  |  |
| Data Size | 64.7K | 68.9K | 59.6K | 378.8K |
| Task Description | Move Arm Hammer Diaper Pail Refills 12 Pack from the Brunel-style bar stool to the white 2-seater sofa. | Move Shark from the Conlay kitchen to the comfortable sofa. | Move wood block from the 7-piece dining set with grey chairs to the Low kitchen element, Natural element. | Move flat screwdriver from the Modern Industrial Dresser, Natural Material to the Stacked shelf system. |
| Context Description | I have embarked on my task and am steadily advancing toward the Brunel-style bar stool, where the Arm Hammer Diaper Pail Refills 12 Pack MFWkmoweejt is situated. | I have embarked on my task and successfully navigated to the Conlay kitchen, retrieving the Shark with ease. Now, I am inching closer to the cozy haven of the comfortable sofa, where I will soon place the Shark. | The task has started and I have navigated to 7-piece dining set with grey chairs and picked up the wood block , I am getting closer to Low kitchen element, Natural element where I should place wood block. | The journey has commenced, and I have successfully navigated to the Modern Industrial Dresser, Natural Material, where I have now picked up the flat screwdriver. |
| Action Information | [[68, 755, 239, 967]] | [[447, 539, 999, 999]] | [[246, 666, 285, 705]] | 4 |

**Stage 1: Task Planning with PDDL**    Using Habitat simulation [26], we first construct symbolic task plans to complete OWMM tasks based on Planning Domain Definition Language (PDDL) [22]. PDDL provides a structured representation of preconditions, actions, and effects, enabling systematic generation of valid task sequences. This ensures that all generated episodes are logically consistent and executable.

**Stage 2: Trajectory Execution and Data Collection**    We direct the robot to execute task sequences within the simulator, recording key information at each step: robot coordinates, current action, object and receptacle positions, and camera extrinsic parameters. To enhance data collection efficiency, we do not initially collect RGB images at every timestep, but rather mark keyframe candidates during execution.

**Stage 3: Keyframe Selection and Filtering**    We apply strategic keyframe selection to ensure data quality. For navigation, we select steps where the target receptacle is visible as start points and robot stopping points as endpoints, sampling waypoints at intervals. For pick and place actions, we select frames where the target object/receptacle is both visible and reachable by the robotic arm.

**Stage 4: Chain-of-Thought Annotation Generation**    We construct structured chain-of-thought annotations based on predefined templates that incorporate: (i) *task instruction reasoning* for decision context; (ii) *perception reasoning* about current observations; (iii) *action decisions* with affordance grounding (bounding boxes for navigation/pick/place, image IDs for scene retrieval); (iv) *summarization* of the current step for historical context. The summarization from each step is systematically incorporated into the "Robot's History" for the next step's question, enabling temporal reasoning.

To enhance linguistic diversity, we use GPT-4o mini to paraphrase the reasoning and summarization components while preserving the structured format and action annotations. This increases robustness to natural language variation without compromising annotation precision.

Finally, we collect scene graph frames for each episode by sampling robot head-view images at task-relevant locations (initial and goal receptacles) and additional random positions for scene coverage, as detailed in Appendix C.1. Additional implementation details of the data synthesis pipeline are provided in Appendix D.

## 4.2 Dataset Analysis

We used 143 scenes from The Habitat Synthetic Scenes Dataset (HSSD) [13] and combined objects from YCB Objects [3] and Google Scanned Objects [6] to create a dataset with 157 unique manipulation objects and 1,471 receptacles from selected scenes. In each scene, objects were randomly placed for the robot to pick and relocate to another receptacle, with 400 episodes sampled per scene.

Following the data synthesis pipeline described in section 4.1, we collected and filtered episodes from each scene, ultimately gathering 21,046 valid episodes with approximately 572K total annotations. As shown in Table 5, the final dataset comprises: i) pick action dataset of 64.7K image-text pairs, ii) place action dataset of 68.9K image-text pairs, iii) navigation dataset of 59.6K image-text pairs, and iv) search scene frame dataset with 378.8K multi-image-text pairs.

In our datasets, we also apply a re-labeling process for objects and receptacles, unlike HomeRobot's fixed criteria[40]. We kept the original object labels and used GPT-4o to rewrite receptacle labels. These labels were diverse and descriptive, suited for open-world scenarios.

## 5 Experiments

In this section, we present the evaluation results in both simulation and real-world data. We present the experimental results of single-step evaluation for *OWMM-VLM* in our simulated benchmark in section 5.1 and episodic evaluation for the *OWMM-Agent* in our simulated benchmark in section 5.2. We then present the real-world evaluation in section 5.3. Due to the page limit, we discuss the data scaling law and how data diversity impacts the model performance in Appendix D.2. For the ablation study on model design, such as the choice of generating bounding boxes rather than points, please see Appendix G. We further provide the qualitative comparisons of different models in Appendix J. Additional analysis including failure mode categorization (Appendix H) and computational efficiency with varying frame counts (Appendix I) are also available in the appendix.

Table 2: **Single-step evaluation of VLM models on OWMM core multi-modal capabilities.** The OWMM-VLM-38B model achieves the best performance across all metrics, demonstrating its superior ability to integrate scene understanding, decision-making, and action generation. *: Since PIVOT and RoboPoint are designed for a single image, we also report the single image grounding results for fairness.

| Model/ Task Score | Ego-centric Decision-making↑ | Image Retrieval↑ | Affordance Grounding (object)↑ | Affordance Grounding (receptacle)↑ | Affordance Grounding (navigation)↑ | Time Consumption(s)↓ |
|---|---|---|---|---|---|---|
| OWMM-VLM-38B(ours) | **97.85%** | **87.54%** | **0.97(±0.14)** | **0.94(±0.19)** | **0.88(±0.17)** | 36.58 |
| OWMM-VLM-8B(ours) | 96.72% | 79.04% | 0.93(±0.14) | 0.91(±0.20) | 0.83(±0.21) | 16.58 |
| GPT-4o[11] | 48.53% | 46.46% | 0.56(±0.38) | 0.35(±0.40) | 0.07(±0.21) | 160.74 |
| Internvl2.5-8B[5] | 17.52% | 1.27% | 0.05(±0.19) | 0.18(±0.31) | 0.14(±0.26) | 16.06 |
| GPT-4o+PIVOT[25] | 52.72% | 55.38% | 0.67(±0.38) | 0.45(±0.44) | 0.05(±0.18) | 22.91 |
| GPT-4o+Robopoint[42] | 49.56% | 49.72% | 0.64(±0.41) | 0.38(±0.42) | 0.06(±0.20) | 14.19 |
| Test of Single Image Grounding(*) | | | | | | |
| Robopoint[42]* | — | — | 0.91(±0.33) | 0.83(±0.11) | 0.72(±0.11) | — |
| PIVOT(GPT-4o)[25]* | — | — | 0.86(±0.13) | 0.84(±0.12) | 0.74(±0.13) | — |

Table 3: **Agent success rate in OWMM Task.** OWMM-VLM-38B model consistently outperforms others across all metrics.

| Method | Full Task | Image Retrieval(Object) | Robot close to Object | Object Picked | Image Retrieval(Goal) | Robot close to Goal | Dead Loop |
|---|---|---|---|---|---|---|---|
| OWMM-VLM-38B(ours) | **21.90%** | **88.56%** | **84.64%** | **38.56%** | **30.39%** | **23.53%** | **0/308** |
| OWMM-VLM-8B (ours) | 9.45% | 81.43% | 74.59% | 17.92% | 15.96% | 10.42% | **0/308** |
| GPT-4o+PIVOT | 0.33% | 59.15% | 10.13% | 0.65% | 0.33% | 0.00% | 195/308 |
| GPT-4o+Robopoint | 0.33% | 56.86% | 11.11% | 1.31% | 0.00% | 0.00% | 184/308 |
| Experiment with more lenient distance tolerance | | | | | | | |
| OWMM-VLM-38B(ours) | **51.52%** | **89.23%** | **88.22%** | **62.96%** | **51.52%** | **44.78%** | **0/308** |
| OWMM-VLM-8B (ours) | 38.59% | 83.22% | 81.21% | 52.35% | 39.93% | 33.56% | **0/308** |
| GPT-4o+PIVOT | 1.68% | 60.27% | 12.12% | 5.39% | 1.68% | 1.35% | 204/308 |
| GPT-4o+Robopoint | 3.03% | 52.86% | 10.10% | 4.04% | 2.69% | 1.35% | 209/308 |

## 5.1 Single-step Evaluation

In the single-step evaluation, we assess three core VLM capabilities for the OWMM task: 1) Egocentric Decision-making: We evaluate the success rate of choosing correct action categories. 2) Image Retrieval: We measure the image retrieval success rate. 3) Affordance Grounding: Instead of predicting points directly like in [42, 25], *OWMM-VLM* outputs a bounding box, from which we compute the center as the target point. With the target point, we compute the score for affordance grounding by $s = \Sigma_i \mathbb{1}_{valid}(i) \times (1 - norm\_dist_i)$, where $\mathbb{1}_{valid}(i)$ is the indicator function of whether the model generates: an action matched with ground truth and a valid bounding box or point on the $i - th$ test case. 1 if both conditions are satisfied simultaneously, and 0 otherwise. $norm\_dist_i \in [0, 1]$ is the distance between the predicted target point and the ground truth point, normalized by the diagonal of the image. In short, $s \in [0, 1]$ measures VLM's ability to generate accurate grounding with the correct format. Higher scores indicate better performance.

Regarding the baseline methods, we have evaluated both 1) multitasking foundation VLM models, including GPT-4o[11] and InternVL-2.5-8B that share the same unified input and output configuration as ours and 2) modularized agent with multiple models, including GPT-4o+PIVOT[25] and GPT-4o+Robopoint[42]. For Robopoint and PIVOT, which specialize in grounding, GPT-4o serves as the higher-level module for decision-making and image retrieval. If GPT-4o's actions need grounding, its outputs are combined with task details as input to Robopoint and PIVOT for grounding.

The results are reported in Table 2. Our model excels in decision-making, achieving state-of-the-art results in image retrieval and affordance grounding. GPT-4o and InternVL2.5, as generalist models, perform poorly at affordance grounding. In contrast, RoboPoint and Pivot that concentrated on affordance grounding, exhibit capabilities on par with our model in this task, indicating that existing specialized approaches already provide good effect on robot's action affordance.

Moreover, our model demonstrates a marked improvement over GPT-4o in decision-making tasks. This advantage directly translates into higher overall accuracy compared to methods that employ GPT-4o as the agent. In other words, using the data from our data synthesis pipeline to conduct a supervised fine-tuning yields a significant enhancement in robotic decision-making performance.

## 5.2 Episodic Evaluation

In episodic evaluation, we assess how well each model completes an OWMM task episode in the simulator. Task success is measured by placing objects in goal receptacles using distance thresholds of 0.85m or 1.7m. The 0.85m threshold relates to half the average diagonal length of goal receptacles' 3D bounding boxes in our test set.

Additionally, we introduce three metrics to assess subgoals: 1) Image retrieval: Success rate in locating object and goal receptacles from multiple posed images. 2) Object Picked: The success rate of the robot grasping an item when its end effector is either within 0.15m or 0.8m of the target, with the latter matching standard HomeRobot setups [40]. 3) Robot close to: The success rate of robot staying within 1.5m or 2.0m of the object or goal receptacle before picking or placing. Additionally, we propose the "dead loop" metric to quantify the number of cyclic stagnations occurring during test episodes. As mentioned in 5.1, GPT-4o may erroneously output image retrieval decisions when the expected action is navigation, thereby inducing cyclic stagnation. Detailed experimental results are presented in Table 3. See Appendix F for extra details about evaluation settings. We provide a comprehensive failure mode analysis categorizing 100 failed episodes in Appendix H.

## 5.3 Real world Evaluation

In our real-robot experiments, we adopted the mobile manipulation system described in Robi Butler[35] within a real-world home environment. For safety reasons, we cannot allow the agent to fully operate the fetch robot in the real world. When OWMM-VLM generates a multi-modal action to execute, the agent prompts the visualization of the action and waits for human confirmation, and the fetch robot only executes the action with human consent.

We first had the robot navigate through the scene with human control to perform SLAM process. We then select 10 test samples from the sequence of the robot's head view during its run. We used human operators to judge the model's output according to several criteria: whether the chosen action was correct, whether the predicted affordance was accurate, and whether the target was reachable, among

other factors. The results of these experiments are presented in Table 4. The results show that the model trained on synthetically generated data in the simulator also demonstrates strong zero-shot generalization capability in real-world scenarios. Table 5 presents the agent action prediction result on real-world data.

Table 4: **Real world single evaluation.** OWMM-VLM-38B model achieved the best performance, and OWMM-VLM-8B model also outperformed the baseline. While the baseline model demonstrated relatively strong affordance grounding capabilities for objects, its poor performance in action decision-making led to incorrect navigation.

| Method | Image Retrieval | Affordance Grounding(object&receptacle) | Affordance Grounding(navigation) | Total Acc |
|---|---|---|---|---|
| OWMM-VLM-38B(ours) | 7/10 | **10/10** | **10/10** | **90.00%** |
| OWMM-VLM-8B (ours) | 5/10 | **10/10** | 9/10 | 80.00% |
| GPT-4o+PIVOT | **8/10** | 6/10 | 0/10 | 46.67% |
| GPT-4o+Robopoint | **8/10** | 6/10 | 0/10 | 46.67% |

Table 5: **Demonstration of single step evaluation in real world.** These demos showcase OWMM-VLM-38B's outputs, illustrating that even though its training data are drawn entirely from our data-synthesis approach in the simulator, the model delivers outstanding decision-making and affordance-grounding performance in real-world settings.

| Model's Output Action | Pick | Place | Nav to point |
|---|---|---|---|
| |  |  |  |
| Task Description | Move the NutriSoy Bean Milk Box from the Minimalist Black Workstation Desk to the White Rectangular Office Meeting Table. | Move the banana from the black desk to the White Rectangular Office Meeting Table. | Move the chip box from Genuine Leather Sofa to the white table. |
| Context Description | The task has started and I am getting closer to the Minimalist Black Workstation Desk where the NutriSoy Bean Milk Box is located. | The task has started and I have navigated to the black desk and picked up the banana, I am getting closer to the White Rectangular Office Meeting Table where I should place the banana. | The task has started and I am getting closer to Genuine Leather Sofa where the chip box is located. |
| Action Information | [576, 263, 769, 548]] | [[0, 445, 1000, 999]] | [[539, 978, 578, 999]] |

## 6   Conclusion

In this paper, we introduced OWMM-Agent, a novel agent architecture featuring the OWMM-VLM, a vision-language model fine-tuned via a simulation-based agentic data synthesis pipeline for Open-World Mobile Manipulation (OWMM) tasks. This approach enables the VLM to learn state tracking, multi-view reasoning, and multi-modal action generation grounded in global scene understanding and agent embodiment. Extensive experiments demonstrated that our OWMM-VLM, particularly the 38B variant, achieves state-of-the-art performance in single-step multi-modal capabilities like egocentric decision-making and affordance grounding, outperforming generalist VLMs and specialized robotics models. Episodic evaluations in simulated environments further confirmed the OWMM-Agent's superior success rates and robustness against common failure modes like dead loops, while real-world tests on a Fetch robot indicated strong zero-shot generalization. Ablation studies underscored the importance of our design choices, such as bounding box prediction and integrated reasoning, and revealed that while data scaling is crucial, egocentric spatial intelligence can be learned effectively even with limited object and scene diversity if data volume is sufficient. Future work will focus on addressing limitations like pre-mapping reliance and enhancing cross-embodiment adaptability for more complex manipulation tasks. Please also refer to the appendix for discussions about the potential impact of this research in Appendix A and extended discussions on limitations in Appendix B.

## Acknowledgements

We sincerely thank Anxing Xiao and David Hsu from the National University of Singapore for their crucial support and guidance in the successful real-world deployment of this project, as well as their generosity in providing the Fetch robot.

For the funding support, this paper is partially supported by the National Key R&D Program of China No.2022ZD0161000.

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

# Appendix

## A Impact Statement

This work contributes to the long-term vision of creating generalist household robots capable of assisting with daily activities in homes and other human-centric spaces. Ethically, deploying such systems raises considerations regarding safety, privacy, and workforce displacement. Ensuring safe interactions with humans and securing data used for training are critical priorities. In addition, while automation may replace certain household jobs, it also creates opportunities for new roles in robot design, deployment, and maintenance. Future societal implications include increased accessibility to robotic assistance for individuals with disabilities or aging populations. By addressing current limitations through continued research into adaptability and real-world robustness, OWMM-VLM can pave the way toward more inclusive and effective robotic solutions for societal benefit.

## B Limitations

In this work, we have proposed a novel embodied agent architecture with a foundational VLM model to address the open-world mobile manipulation problem. However, we also identify some limitations of our approach.

**Pre-mapping:** Although our method does not require 3D reconstruction of the environment, we still assume a pre-mapping phase with a camera pose graph and 2D occupancy map for path planning in navigation.

**Complex manipulation:** Following the grasping setup in [40], our agent and model can be directly applied robot with suction as end effector. However, our model fells short in the circumstances when the robot needs to control complex end effectors like dexhands.

**Cross-embodiment:** As demonstrated in the experiments, our model learns the object-scale prior for spatial understanding and reasoning. However, when deploying the model onto other robots with different mechanical compositions such as maximum arm stretch distance, our model could fail, i.e. the cross-embodiment issue.

## C Implementation Details

Regarding the model's architecture, we have trained two variants consisting of 8 billion and 38 billion parameters, based on the pre-trained model from InternVL-2.5[5]. The 8B model is composed of InternViT-300M and InternLM-2.5-7B[2], and the 38B model is composed of InternViT-6B and Qwen2.5[38]. We directly finetune the base model on our OWMM dataset. The OWMM-VLM model is trained to autoregressively generate the response tokens consisting of the output action and its corresponding task context in JSON format. Specifically, we freeze the parameters in ViT and only adjust the parameters in MLP and LLM. As for the training time, *OWMM-VLM-8B* is trained on 8X NVIDIA A100 GPUs for about 7 hours, and *OWMM-VLM-38B* is trained on 24X NVIDIA A100 GPUs for about 18 hours. Both our models were trained for 1 epoch. For the testing, we deploy *OWMM-VLM* and RoboPoint[42] locally and use the openAI API to access GPT-4o and PIVOT[25].

### C.1 Pre-mapping

This section provides comprehensive implementation details for the pre-mapping stage and camera pose selection, which are essential for reproducing our approach.

**Simulation Environment** In simulated experiments, we utilize the navigation mesh provided by Habitat environment, where navigable locations are represented as mesh triangles. This pre-computed

navmesh serves as a map for localization and navigation, similar to game engines. The navmesh provides collision-free navigation space and enables efficient path planning for the mobile base.

**Real-World Deployment**   In the lab environment, we run RobiButler [35] over Fetch Lidar input for the pre-mapping stage and localize the robot geometrically. More specifically, RobiButler uses Gmapping [9] algorithm to compute the 2D occupancy map from lidar data. The map is further used to localize the robot and camera poses during task execution. This approach is practical and widely adopted in commercial robotic systems such as robotic vacuum cleaners.

### C.2   Camera Pose Selection

Considering the context length of current VLMs, we select 8 views from the pre-mapping stage plus 1 current egocentric view in our experiments, as demonstrated in Figure 2. This design balances scene coverage with computational efficiency.

**Simulation**   As detailed in Section 4.1, we employ a strategic sampling approach:

1. First, we position the robot at the location of the receptacle where objects were initially located and at the goal receptacle, sampling the robot's head-view images. This ensures coverage of task-critical locations.

2. Subsequently, we randomly position the robot and capture its head-view images to ensure sufficient scene coverage while maintaining computational efficiency.

3. This sampling strategy guarantees that both the initial and goal regions are represented in the pose graph while providing diverse viewpoints of the environment.

**Real Robot Deployment**   In real robot experiments, we extract keyframes from the ROS bag RGB data and manually select the frames related to the task to guarantee enough scene coverage. This manual selection serves as a proxy for a ground-truth frame selection mechanism.

We acknowledge that autonomous and online frame selection from videos are critical for full autonomy but is beyond the scope of the current paper. This could represent a promising direction for future work to enable fully autonomous mobile manipulation in dynamic environments.

## D   Details of Datasets

sectionDetails of Datasets

### D.1   Extra Dataset Construction Details

Our evaluation pipeline is constructed using the HomeRobot [40] framework, which serves as a software structure designed to enable comprehensive benchmarking in both simulated and real-world settings. Specifically, we use the simulation part of HomeRobot project, built on Habitat platform [26], with 200 scenes, 150 categories, and 7,892 object instances.

The original episodic data in HomeRobot are generated with Stretch Robot [12], which has a special telescopic arm instead of a normal articulated arm with rotary joints. This adds additional difficulty in base control as it requires the mobile chassis to rotate accurately to align the arm with the target object for manipulation. However, the baseline VLMs and methods we are going to compare with are designed for robots with conventional articulated arms [42, 25], providing a broad range of chassis poses that allow for successful arm manipulation.

Therefore, we recreate the OWMM episodic training and testing datasets in the simulation using the Fetch Robot, which is a mobile robot equipped with a standard articulated arm and has also been integrated into the Habitat platform.

We partitioned the scenes into training and testing sets using a ratio of 113:30. Besides, we allocated 157 objects between the training and validation sets with a ratio of 137:20, ensuring that the testing set contained entirely unseen objects. This division resulted in a total of 152k training data entries and 4k testing data entries, establishing a robust dataset for training and testing in our OWMM task.

**Keyframe Sampling Strategy**    In the dataset construction pipeline, we first sample key information at each step. This information included the robot's coordinates, current action, the positions of objects and receptacles, and the extrinsic parameters of the robot's head-view camera. In particular, at this stage, we did not collect the robot's head-view images to enhance the data collection efficiency. We recollected the robot's head-view images of these steps within the simulator after applying the selection strategy.

For navigation actions, among all steps that the robot is moving, we select the step that the receptacle is visible from the robot's head-view image as the start point of the navigation action. The point at which the robot stops moving is considered the end point of the navigation action. Within these steps, we sample the waypoint step data at specified intervals.

For grasp and manipulation actions, we select the first three frames during which the robot executes the action as the pre-defined action data.

**Data Filtering Pipeline**    The data filtering pipeline ensures the following criteria:

- For navigation actions, both the receptacle and the next waypoint are within the robot's head-view image.
- For grasp actions, the object to be grasped is reachable by the robotic arm, and the object is within the robot's head-view image.
- For manipulation actions, the receptacle intended for object placement is reachable by the robotic arm, and the receptacle is within the robot's head-view image.

After filtering, approximately 10% of initially collected frames are rejected, ensuring high annotation quality.

**Linguistic Diversity Enhancement**    To enhance the diversity of the dataset, we paraphrased reasoning and summarization parts of the answers using GPT-4o mini, while keeping the action annotations and affordance bounding boxes unchanged. This process increases linguistic diversity by generating 3-5 paraphrases per template, resulting in more robust language understanding without additional manual annotation effort.

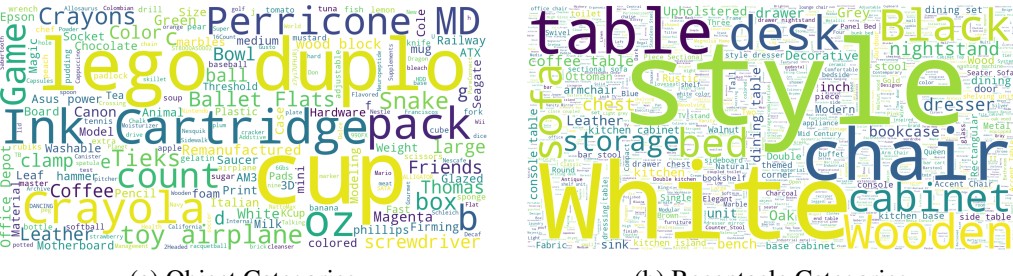

(a) Object Categories                          (b) Receptacle Categories

Figure 4: **Word Cloud Distribution of Objects and Receptacles in our dataset.** Object categories show diverse household items (tools, food containers, toys), while receptacles include varied furniture types (tables, shelves, cabinets) with rich descriptive labels.

### D.2    Analysis on the training data

This analysis tries to answer two questions: 1) **How does the diversity of objects and environments affect the model's performance on unseen objects and environments in the test set**? We examine dataset diversity using three 45k-sample sets: 100% scenes and objects, 100% scenes with 30% objects, and 30% scenes with 100% objects. We control the total number of training samples while changing the number of object instances or scenes appearing in the training data. 2) **How does the model's performance change as the training data scales up?** For data scaling, we use five data sizes: 0k (no fine-tuning), 15k (10%), 45k (30%), 76k (50%), and 152k (100%). At 0k, we give the Internvl-2.5-8B model limited input-output pairs, allowing it to generate structured outputs via

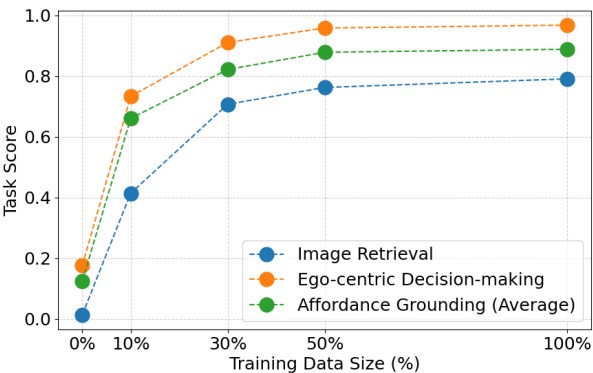

Figure 5: **OVMM-VLM-8B Sub-task Performance with the Increase of Training Data Size.** The task scores consistently improve as the training data size increases.

in-context learning. We evaluate the performance in image retrieval, egocentric decision-making, and three affordance grounding subtasks. Results are shown in Table 6 and Figure 5.

The results for the first question show that object and scene diversity have negligible effects on multi-modal capabilities, as metric fluctuations remain within a $5\%$ range. For the second quesion, data scaling is crucial for enhancing OWMM-VLM's performance. As seen in Figure 5, increasing the dataset from 0k to 152k samples shows a logarithmic improvement, especially at lower sizes (0k to 15k, and 15k to 45k). However, benefits diminish near 152k. While larger datasets aid generalization, marginal gains decrease beyond a threshold. As performance gains plateau, egocentric decision making approaches a success rate of 1.0, whereas image retrieval lingers at approximately 0.8. This difference is likely due to the model's limited capacity with 8 billion parameters. We also draw two extra observations from the experiment:

1) The embodiment prior for deciding the current action based on the ego-centric RGB image, especially how close the robot should be to interact with the target objects, can be learned in a data-driven approach.

2) The ability to comprehend multiple images or the multimodal context length may present one of the bottlenecks for VLM models to function as the core cognitive model for intelligent robots, particularly when scene-level understanding is essential.

Table 6: **Results with different data diversity data scales.** The best performance across training sets with different scales is indicated with **bold font**. Besides, underline highlights the best performance across three 45k-sample training sets with different diversity.

| Data Composition/ Task Score | Ego-centric Decision-making↑ | Image Retrieval↑ | Affordance Grounding (object)↑ | Affordance Grounding (receptacle)↑ | Affordance Grounding (navigation)↑ |
|---|---|---|---|---|---|
| 0k(0%) | 17.52% | 1.27% | 0.05($\pm$0.19) | 0.18($\pm$0.31) | 0.14($\pm$0.26) |
| 15k(10%) | 73.27% | 41.36% | 0.69($\pm$0.43) | 0.84($\pm$0.29) | 0.45($\pm$0.41) |
| 45k(30%) | 91.01% | 70.68% | 0.88($\pm$0.24) | 0.84($\pm$0.31) | 0.74($\pm$0.31) |
| 45k(100% scene + 30% object) | 91.56% | 71.95% | 0.87($\pm$0.26) | 0.89($\pm$0.23) | 0.72($\pm$0.33) |
| 45k(30% scene + 100% object) | 88.96% | 69.12% | 0.87($\pm$0.26) | 0.84($\pm$0.31) | 0.69($\pm$0.36) |
| 76k(50%) | 95.79% | 76.20% | 0.91($\pm$0.19) | 0.88($\pm$0.24) | **0.84($\pm$0.20)** |
| 152k(100%) | **96.72%** | **79.04%** | **0.93($\pm$0.14)** | **0.91($\pm$0.20)** | 0.83($\pm$0.21) |

# E   Details of Baseline Setting

As Robopoint and PIVOT are designed for single-image QA task, we adjusted some settings to enable them fully utilizing their capabilities under the OWMM task.

### E.1 Single Image Grounding

For the single-step evaluation, we first extracted robot's task instruction from the original prompt of the current step. Based on the ground truth action of the current step and whether the robot picks up an object, we designed new task instructions, as shown in Table 7. For Robopoint, we appended the

Table 7: **Redefined Task Instructions.** {object item}, {target rec} and {goal rec} are from robot's task instruction.

| Ground Truth Action | New Task Instruction |
|---|---|
| Pick | The robot needs to pick {object item} on {target rec} |
| Place | The robot needs to place {object item} on {goal rec} |
| Nav to point(object picked) | The robot needs to navigate closer to the {goal rec} for placing {object item} |
| Nav to point(object not picked) | The robot needs to navigate closer to the {target rec} for picking {object item} |

following context: "Find a few spots for robot to execute the action. Your answer should be formatted as a list of tuples, i.e. [(x1, y1), (x2, y2), ...], where each tuple contains the x and y coordinates of a point satisfying the conditions above. The coordinates should be between 0 and 1, indicating the normalized pixel locations of the points in the image." This configuration aligns with Robopoint's original settings.

For PIVOT, we configured the following parameters: n_samples_init=10, n_samples_opt=6, n_iters=2. In our evaluation settings, as the input consists of a single RGB image and task instructions, we randomly sample initial points in the image from a 2D Gaussian distribution. The distribution is parameterized with a mean of (256, 256) and standard deviation of (100, 100).

### E.2 Agent Setting

We employed GPT-4o for agent construction.GPT-4o first receives our instruction inputs and returns JSON-formatted responses. When gpt's output action is "search scene frame", we directly adopt GPT-4o's response as the agent's current-step output. For actions "nav to point", "pick", or "place", the system sends both the action name and robot's current-view RGB image (single frame) to Robopoint/PIVOT for action affordance. The reformulated task instruction sent to Robopoint/PIVOT follows this template:

"The robot needs to {task_instruction}. Now the robot needs to {gpt_output_action}. {robot_history}"

where {task_instruction} is the original task instruction,{gpt_output_action} is gpt's output action,{robot_history} is the summarization of previous step. In single-step evaluation, Robopoint and PIVOT process these new task instructions using the same methodology described in Appendix E.1.In episodic evaluation, we transmit depth information to PIVOT while maintaining consistency with its original configuration.

## F   Extra Details of Episodic Evaluation

As mentioned in section 5.2, we designed the following metrics for episodic evaluation. More detailed specifications of these metrics are outlined below:

Object to Goal Distance: We used the object to goal distance as the metric to determine whether objects are successfully placed in goal receptacles. To establish appropriate thresholds, we first calculated the 3D bounding box diagonal distances of all goal receptacles in the test set, filtering out those with distances less than 0.75m or greater than 3m.Table 8 show some examples of goal receptacles in our test set.Subsequently, we computed the average diagonal distance (1.7m) from the remaining valid receptacles. Based on this value, we selected half of the average (0.85m) as the strict threshold criterion and the full average (1.7m) as the relaxed threshold criterion. This threshold approach ensures successful placement recognition when robots position objects near goal receptacles,and reasonable constraint boundaries to prevent excessive leniency in evaluation.

Table 8: **Example Goal Receptacles in our Test Set.** The numbers in the figure represent the diagonal distances of the 3D bounding boxes of the receptacles. This indicates that there is significant variation in the sizes of the goal receptacles in the test set.

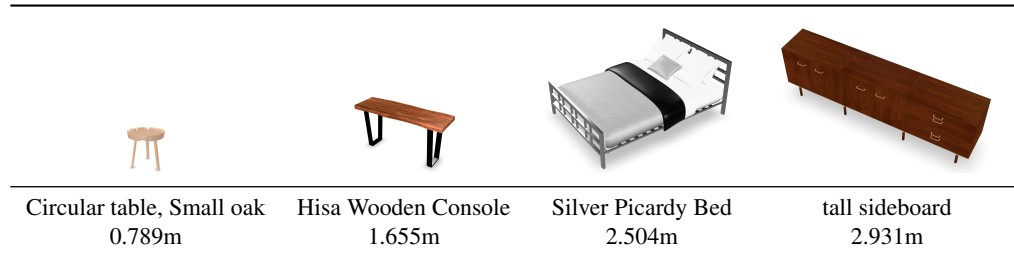

| Circular table, Small oak | Hisa Wooden Console | Silver Picardy Bed | tall sideboard |
|:---:|:---:|:---:|:---:|
| 0.789m | 1.655m | 2.504m | 2.931m |

## F.1 Simulation

For one simulation step, the robot state delta is calculated by forward kinematics, as implemented by the Habitat 3.0 environment[26]. The robot state and observations updates can be expressed mathematically as:

$$\mathbf{x}_{t+1} = f_k(\mathbf{x}_t, \mathbf{a}_t, \Delta t)$$
$$I^c_{t+1}, D^c_{t+1} = f_{obs}(\mathbf{x}_{t+1})$$

where $\mathbf{x}_t$ stands for robot current state (e.g., joint angles, positions), $\mathbf{a}_t$ stands for velocities, and $f_k$ represents the kinematic model function that computes the next robot state within the discretized time step of duration $\Delta t$. $f_{obs}$ is the observation model function, decided by the sensor link forward kinematics function and camera model.

## G Ablation Study on OWMM-VLM

The ablation study evaluates the contributions of the components of the OWMM-VLM model. We focus on grounding output formats, comparing the bounding box and point coordinate, and we assess the inclusion of reasoning and summarization in the outputs. Furthermore, we examine the beam search option provided by the base model Internvl-2.5-8B[5]. The results are in Table 9.

From the table, we have these observations and indications:

1) **Beam Search.** Beam search is a decoding algorithm widely used in language generation, maintaining a beam number of top candidate sequences at each step. Beam search enhances Ego-centric Decision-making and Affordance Grounding tasks, with minimal impact on Image Retrieval, but increases temporal and spatial overhead in inference, especially on the 38B variant. Hence, its effect is briefly shown only in the ablation study.

2) **Grounding Format.** Replacing bounding box predictions with direct output coordinates reduces performance in Affordance Grounding, especially for objects ($0.9251 \rightarrow 0.6542$) and receptacles ($0.9060 \rightarrow 0.6479$). It is postulated that the large-scale visual grounding data in the pre-trained model allow our model to utilize this prior knowledge. The consistency in output format between the base model and the instruction fine-tuning dataset aids the training process.

3) **Reasoning and Summarization**. Removing reasoning and summarization capabilities leads to the worst performance across most metrics, with a decrease in Image Retrieval ($0.7904 \rightarrow 0.6586$) and Ego-centric Decision-making ($0.9672 \rightarrow 0.9049$). This highlights the critical role of reasoning and summarization in maintaining contextual coherence and task understanding.

Table 9: **Ablation Study on OWMM-VLM.** The best performance is indicated with **bold font**.

| Model Ablation | Ego-centric Decision-making↑ | Image Retrieval↑ | Affordance Grounding (object)↑ | Affordance Grounding (receptacle)↑ | Affordance Grounding (navigation)↑ |
|---|---|---|---|---|---|
| OWMM-VLM-8B | 96.72% | **79.04%** | 0.93($\pm$0.14) | 0.91($\pm$0.20) | 0.83($\pm$0.21) |
| + beam search | **97.30%** | 78.47% | **0.97($\pm$0.14)** | **0.93($\pm$0.21)** | **0.85($\pm$0.18)** |
| + output-coord | 96.70% | 78.19% | 0.65($\pm$0.15) | 0.65($\pm$0.17) | 0.63($\pm$0.14) |
| - reasoning and summarization | 90.49% | 65.86% | 0.88($\pm$0.24) | 0.83($\pm$0.33) | 0.82($\pm$0.24) |

# H Failure Mode Analysis

To better understand the limitations and bottlenecks of our system, we conducted a comprehensive failure analysis on 100 randomly selected failed episodes from the evaluation logs. Since the current evaluation pipeline does not support automatic failure case analysis, we manually reviewed the action sequences and categorized failures into four distinct types:

## H.1 Failure Categories

**Ego-centric Decision Making Error (36%)**  These failures occur when the model makes incorrect pick/place or navigation action decisions. Typical scenarios include:

- The robot is close enough to the target but outputs a navigation action, leading it to move away and lose the target from view.
- The robot is not close enough to interact but outputs a pick or place action, resulting in failed execution.
- The model selects an inappropriate action type given the current spatial relationship between the robot and target objects.

**Image Retrieval Error (28%)**  The retrieved image ID is incorrect, meaning the model selects a posed frame that does not contain the target object or receptacle. This can result from:

- Confusion between visually similar objects or receptacles.
- Incorrect interpretation of the language instruction.
- Hallucination when reasoning over multiple candidate images.

**Affordance Grounding (Object/Receptacle) Error (15%)**  The robot is positioned close enough to the target, but the center point of the output bounding box does not intersect with the target object/receptacle, causing a failed pick or place action. Alternatively, the grounded object/receptacle is incorrect despite correct spatial positioning.

**Affordance Grounding (Navigation) Error (21%)**  The output bounding box for navigation corresponds to an unreachable region (e.g., blocked by obstacles or outside the navigable space), or causes the target object/receptacle to disappear from the robot's egocentric view after navigation completes.

From this analysis, we draw two key conclusions:

1. **Bottleneck in object picking/placing actions**: While our model achieves strong performance in early stages (88.56% on object image retrieval, 84.64% on robot navigation to object), performance degrades significantly in later stages (38.56% object picking, 30.39% goal image retrieval). This suggests that manipulation affordance grounding remains a critical challenge.

2. **Error accumulation across subtasks**: Failed early actions compound, making later subtasks increasingly difficult. This cascading effect accounts for the performance gap between single-step and episodic evaluations, as errors in object retrieval or navigation prevent the robot from ever reaching the manipulation stage.

# I Computational Efficiency Analysis

To address concerns about real-world deployment efficiency and scalability to large scenes, we conducted experiments evaluating GPU memory consumption and inference time with varying numbers of input frames.

We evaluated two model configurations:

- **OWMM-VLM-8B**: Single-step inference on a single A100-40G GPU with 2+1, 4+1, 8+1, and 16+1 frames (posed frames + egocentric frame). The result is shown in Table 10.
- **OWMM-VLM-38B**: Single-step inference on 4×A100-40G GPUs with 8+1, 16+1, 32+1, and 64+1 frames using parallel inference. The result is shown in Table 11.

Table 10: OWMM-VLM-8B inference performance with varying frame counts on single A100-40G GPU.

| Input Frames | Prompt Tokens | Time (s) | Memory (GB) |
|---|---|---|---|
| 2+1 | 3409.19 | 3.66 | 17.44 |
| 4+1 | 6057.19 | 3.67 | 17.77 |
| 8+1 (default) | 11353.19 | 4.84 | 18.43 |
| 16+1 | 21945.19 | 7.09 | 19.75 |

Table 11: OWMM-VLM-38B inference performance with varying frame counts on $4\times$A100-40G GPUs with parallel inference.

| Input Frames | Prompt Tokens | Time (s) | Memory (GB) |
|---|---|---|---|
| 8+1 (default) | 2810.37 | 4.39 | 98.22 |
| 16+1 | 4922.37 | 5.04 | 99.13 |
| 32+1 | 9146.37 | 7.34 | 100.65 |
| 64+1 | 17594.37 | 15.33 | 104.30 |

**Scalability** Both models demonstrate reasonable scalability: inference time grows sub-linearly with the number of frames due to efficient multi-image processing in modern VLMs. For the 8B model, doubling frames from 8+1 to 16+1 increases inference time by only 46% (4.84s to 7.09s).

**Memory Efficiency** Memory consumption grows modestly with additional frames, as the visual tokens are processed through the same encoder. The 8B model remains under 20GB even with 16+1 frames, while the 38B model stays around 100GB across all tested configurations.

**Potential Optimization for Real-time Deployment** While these timings reflect comprehensive processing (multi-image processing, chain-of-thought reasoning, and action generation), real-time deployment could benefit from optimization techniques:

- **Image token compression** [16]: Methods that select text-relevant image patch tokens can significantly reduce input token counts.
- **Model quantization** [36]: Quantization-aware approaches can accelerate inference while maintaining performance.
- **Dynamic frame selection**: As discussed in Appendix C.1, adaptive frame selection can maintain scene coverage while controlling computational costs.

# J  Qualitative Evaluation

We provide the qualitative evaluation of our *OWMM-VLM* model compared to other baseline models.

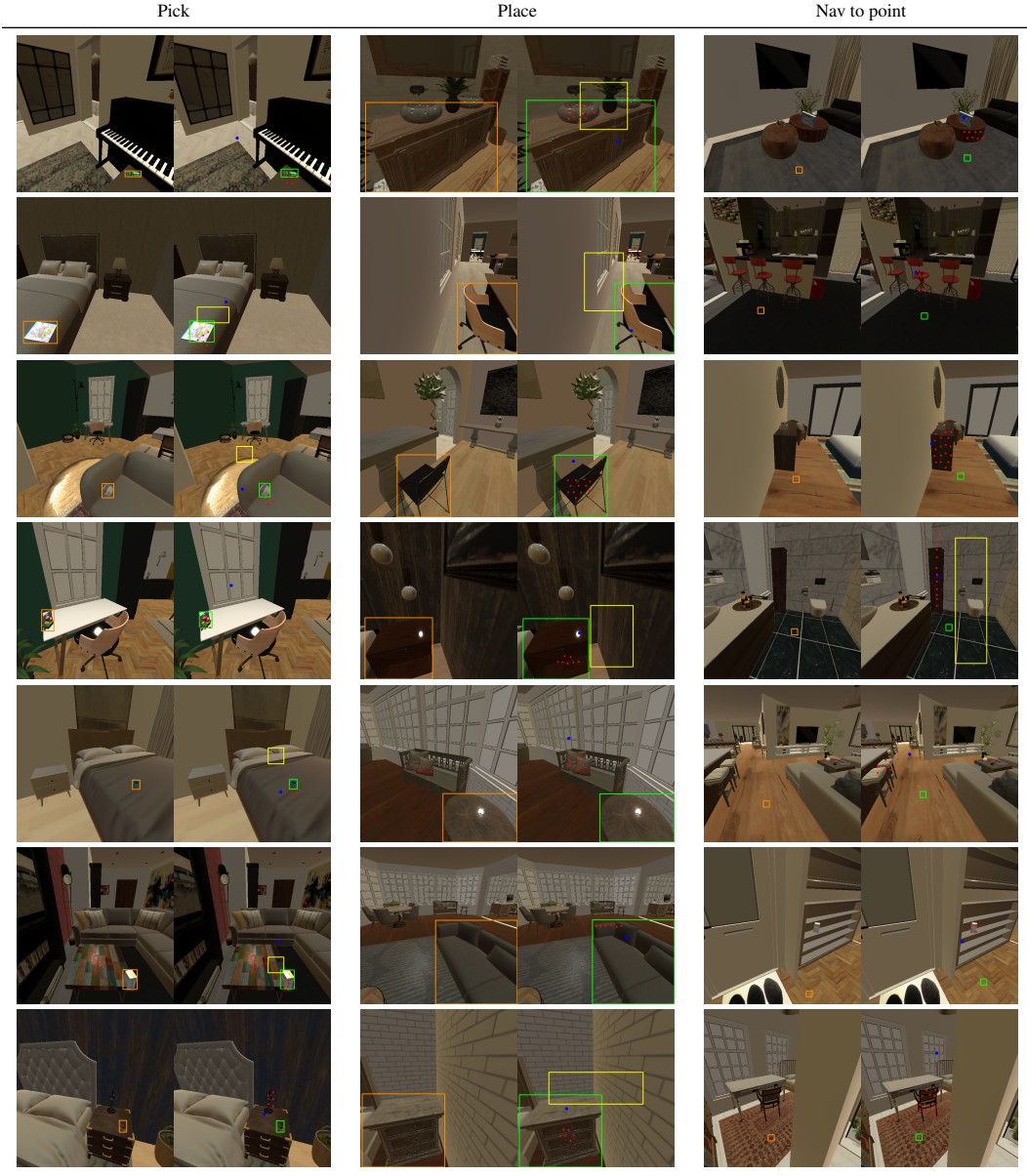

Table 12: **Single step Qualitative Evaluation.** The table demonstrates the single step qualitative evaluation results: — represent the **ground truth**; — represent GPT-4o; ● represents RoboPoint; ● represents PIVOT; — represents InternVL base model; — represent **ours**

| Task Description | Context Description | Response |
|---|---|---|
| Move Schleich Allosaurus from the Upholstered Sofa to the Brown and Gold Accent Cabinet. | Task just started. |  |
| Move Tena Pads Heavy Long 42 pads from the Dark Wooden Tall Open Bathroom Cabinet to the multifunctional games table. | The task has started and I have navigated to Dark Wooden Tall Open Bathroom Cabinet and picked up the Tena Pads Heavy Long 42 pads. |  |
| Move hammer from the Multiple Drawer Short Boy to the Dark Wooden Tall Open Bathroom Cabinet. | The task has started and I have navigated to Multiple Drawer Short Boy and picked up the hammer. |  |
| Move 065-b cups from the Unch metal and wood bar stool to the Magnolia Home Foundry Console Table. | The task has started and I have navigated to Unch metal and wood bar stool and picked up the 065-b cups. |  |

Table 13: **Single step Qualitative Evaluation Search Scene Frame.** The table demonstrates the single step qualitative evaluation search scene frame results: — represent the **ground truth**; — represent GPT-4o; — represents InternVL base model; — represent **ours**

