# OpenReview forum: "OWMM-Agent: Open World Mobile Manipulation With Multi-modal Agentic Data Synthesis"
_NeurIPS.cc/2025/Conference — NeurIPS 2025 poster_

### Official Review · Reviewer_v5R9 · 2025-06-25

**Clarity:** 3
**Significance:** 2
**Originality:** 2
**Rating:** 3
**Confidence:** 4

**Summary:**

This paper proposes OWMM-Agent, a vision-language agent for open-world mobile manipulation. The model, OWMM-VLM, is fine-tuned on a large synthetic dataset and performs high-level action planning (e.g., pick, place, navigate) based on natural language, multi-view images, and robot state history. It interfaces with classical planners for execution. Experimental results show that the proposed method significantly outperforms proposed baselines.

**Questions:**

1. Could you elaborate on how the scene is discretized and how the pose graph is constructed during the pre-mapping phase? Specifically, what strategy do you use to select camera poses and ensure sufficient scene coverage?

2. How does your setting differ from goal-oriented vision-language navigation tasks such as REVERIE [1] and SOON [2]? Since your method has access to pre-captured global scene information (i.e., posed frames), would it be fair to consider your setup a simplified variant of those tasks?

3. If I understand correctly, the pre-captured scene-level posed frames are included as part of the model's input. Could you clarify how the number of these frames affects GPU memory consumption and inference time? In the case of large or complex scenes, would this design lead to significant computational overhead?

4. You mention the method can handle dynamic environments. Could you provide any design details or experimental results to support this claim?

5. What types of visual hallucination occur in this navigation task when using foundation models, and how does your method address them?

6. Writting issue:
    - Line 217, Table 5 => Table 1

[1] Qi, Y., Wu, Q., Anderson, P., Wang, X., Wang, W. Y., Shen, C., & Hengel, A. V. D. (2020). Reverie: Remote embodied visual referring expression in real indoor environments. In *Proceedings of the IEEE/CVF Conference on Computer Vision and Pattern Recognition* (pp. 9982-9991).

[2] Zhu, F., Liang, X., Zhu, Y., Yu, Q., Chang, X., & Liang, X. (2021). Soon: Scenario oriented object navigation with graph-based exploration. In *Proceedings of the IEEE/CVF Conference on Computer Vision and Pattern Recognition* (pp. 12689-12699).

**Ethical Concerns:**

["NO or VERY MINOR ethics concerns only"]

**Final Justification:**

Several concerns were addressed, but key issues remain.

1. The paper’s distinction from existing VLN work is unclear.
2. The claimed link between visual hallucination and synthetic data design is weak. The method design is overly intuitive and trivial.

I will maintain my score and defer these concerns to the AC for thorough consideration.

**Limitations:**

yes

**Paper Formatting Concerns:**

no concerns

**Quality:**

2

**Strengths And Weaknesses:**

Strengths:

1. Propose a unified vision-language agent to support multimodal training
2. Introduces a scalable synthetic data generation pipeline covering four grounding tasks (pick, place, navigation, and scene search).
3. Both simulation-based evaluations and real-world zero-shot tests

Weaknesses:

1. The number of real-world test cases is very limited, which undermines the statistical significance of the reported results.
2. The model design may introduce significant computational bottlenecks, raising concerns for real-world deployment efficiency.
3. Some claims lack sufficient clarification (see questions)

---

> ### Author Rebuttal · Authors · 2025-07-29
>
> We really thank the reviewer for the time and efforts invested in this thoughtful evaluation of our paper. We would also appreciate the recognition of our work's strengths, particularly the acknowledgment of the unified agent framework and data synthesis pipeline. Apart from the positive comments, we would like to address the concerns and questions raised by the reviewer:
> ***
> ## Response to Weaknesses
>
> ### Limited Real-World Test Cases
>
> Our real-world experiments, conducted in from a single lab environment, were designed as a proof-of-concept to demonstrate two ideas: 1) the proposed embodied **agent architecture** can be deployed onto the real mobile manipulation robot in a reasonable way; 2) the fine-tuned **VLM models** from synthesized dataset has zero-shot generalization capability  from simulation to reality.
> ***
> ### Computational bottlenecks and real-world deployment efficiency
>
> While these timings reflect comprehensive processing including multi-image processing, chain-of-thought reasoning and action generation, we acknowledge that real-time deployment requires optimization. Although the optimization of VLM inference is beyond the scope, we provide some analysis of this bottleneck and possible solutions. One clear bottleneck we observe is that whole-scene understanding involves the tokenization of multiple images in the scene, which composes most of the input tokens to VLM model. The image token compression approaches could alleviate this issue. For example, [1] proposed a method to compress image tokens by selecting the text-relevant image patch tokens. Besides, applying model quantization to VLM models could also accelerate inference efficiency, such as [2].
>
> [1] Li, Kevin Y., et al. "Inference Optimal VLMs Need Fewer Visual Tokens and More Parameters." arXiv  (2024).
>
> [2] Xie, Jingjing, et al. "Advancing multimodal large language models with quantization-aware scale learning for efficient adaptation." ACM MM. 2024.
> ***
> ## Answer to Questions
>
> ### Scene Discretization and Pose Graph Construction
>
> We really thank the reviewer for pointing out this question. We also think this is a key concern in real-world system design. We have also acknowledged the pre-mapping stage as one of our main limitations in **Appendix B Limitations** (line 455), specifically, "... we **assume a pre-mapping phase** with a camera pose graph and 2D occupancy map for path planning in navigation ...".
>
> **Pose Graph Construction in Pre-mapping Stage**: In simulated experiments, we get the pre-mappming using the navigation mesh[3] provided by Habitat environment, where navigable locations are represented as mesh triangles. In the lab environment, we run RobiButler[4] over Fetch Lidar input for the pre-mapping stage and localize the robot geometrically.
>
> **Camera Pose Selection**: Considering the context length of current VLMs, we select 8 views from pre-mapping stage and 1 current ego-centric view in our experiments, as demonstrated in Figure 2. In simulation, as detailed in **Section 4.1 line 205**, we "first set the robot at the location of the receptacle where objects were initially located and the goal receptacle, sampling the robot's head-view images. Subsequently, we randomly positioned the robot and captured its head-view images." This ensures sufficient scene coverage while maintaining computational efficiency. In the real robot, we extract the keyframes from the ROS bag RGB data and manually select the frames related to the task to guarantee the scene coverage.
>
> Besides, although actively selecting related frames from videos is beyond the contributions of this paper, there are some related works trying to address this issue. VideoAgent[5] and the follow-up works proposed to perform video understanding by retrieving task-relavant frames and then conducting a normal VLM reasoning process. Here we assume we have a ground truth VideoAgent and focus on downstream tasks.
>
> We acknowledge this is one of the limitations of our method and we plan to add more discussions to the final version of our paper to provide more insight. Addressing this limitation could be future work.
>
> [3] Arkin, Ronald C. "Path planning for a vision-based autonomous robot." *Mobile robots I*. Vol. 727. SPIE, 1987.
>
> [4] Xiao, Anxing, et al. "Robi Butler: Multimodal Remote Interaction with a Household Robot Assistant." ICRA 2025.
>
> [5] Wang, Xiaohan, et al. "Videoagent: Long-form video understanding with large language model as agent." ECCV 2024.
> ***
> ### Comparison with REVERIE and SOON
>
> We thank the reviewer for the question about the positioning of our paper. Our setting differs fundamentally from REVERIE and SOON in several key aspects:
>
> Overall, these two papers can be categorized to Visual Language Navigation tasks. In comparison, this paper focuses on Open World/ Vocabulary Mobile Manipulation as introduced in **Section 2 Related Works (line 77)**. Specifically, our approach differs in three major ways : 1) **Exploration vs. VLM Reasoning on Pre-mapping**: Unlike REVERIE/SOON which focuses on more efficient online exploration, our agent focuses on task-relevant frame retrieval and action decision making based on pre-mapping information; 2) **Manipulation Integration**: Unlike REVERIE/SOON, We address both navigation and manipulation in a unified framework 3) **Low-level Action Predicion**: REVERIE/SOON uses pre-defined view points as navigation action space. This action is not flexible under ego-centric agent, especially for mobile manipulation, in which task the robot needs to carefully choose a navigation point not too distant or too close so that the object falls into the arm workspace. In comparison, our agent can choose a viewpoint to navigate to and can also **generate finer action including moving to any point or grasping any region in ego-centric space**. You can refer to **Table 1 (line 183)** for more details.
>
> ***
> ### How does number of frames affects GPU consumption and inference time?
>
> We thank the reviewer for this question that is important for real-world deployment. We managed to add the new experiment to test the GPU consumption and inference time on two settings:
>
> 1) Evaluation of single-step inference of OWMM-VLM-8B on single A100 GPU with [2+1; 4+1; 8+1; 16+1] frames as image input.
> 2) Evaluation of single-step inference of OWMM-VLM-32B on 4 X A100 GPU with [8+1; 16+1; 32+1; 64+1] frames as image input. Since the larger VLM model cannot be deployed on a single A100-40G GPU, we use parallel inference on 4 GPUs, which is common in VLM  deployment.
>
> The result is below:
>
> | Input Frames | 2+1 |  |  | 4+1 |  |  | 8+1 (default) |  |  | 16+1 |  |  |
> | --- | --- | --- | --- | --- | --- | --- | --- | --- | --- | --- | --- | --- |
> | Model/ Setting | Prompt token | Time avg(s) | Memory | Prompt token | Time avg(s) | Memory | Prompt token | Time avg(s) | Memory | Prompt token | Time avg(s) | Memory |
> | OWMMVLM-8B | 3409.19 | 3.6573 | 17+0.44=17.44G | 6057.19 | 3.6723 | 17+0.77=17.77G | 11353.19 | 4.8412 | 17+1.43=18.43G | 21945.19 | 7.0883 | 17+2.75=19.75G |
>
> | Input Frames | 8+1 (default) |  |  | 16+1 |  |  | 32+1 |  |  | 64+1 |  |  |
> | --- | --- | --- | --- | --- | --- | --- | --- | --- | --- | --- | --- | --- |
> | Model/ Setting | Prompt token | Time avg(s) | Memory | Prompt token | Time avg(s) | Memory | Prompt token | Time avg(s) | Memory | Prompt token | Time avg(s) | Memory |
> | OWMMVLM-38B | 2810.37 | 4.394 | 97+1.22=98.22G | 4922.37 | 5.0403 | 97+2.13=99.13G | 9146.37 | 7.3397 | 97+3.65=100.65G | 17594.37 | 15.3297 | 97+7.30=104.30G |
>
>
> I would like to put more analysis here, but we are **reaching character limitations** with the rest of the questions. Besides, in the case of large or complex scenes, we would propose to use the dynamic frame selection scheme as we discussed in answer to question 1. We think this question is meaningful and we would like to add these new experiments and analysis to the appendix in the final version of this paper.
> ***
> ### Dynamic Environment Handling
>
> We thank the reviewer for this insightful question, though we do not claim "the method can handle the dynamic environments". Instead, we guess that the closest statement in the paper could be "Additionally, they often require time-consuming dense 3D reconstruction, **making them less suitable for complex, open-ended and dynamic environments**" in line 32.
>
> Compared to the 3D reconstruction-based methods, we believe our proposed scene representation of a graph of posed images better suits dynamic scenarios. For example, in a human-robot cohabiting environment, where the scene is changed by the human subject, the robot agent can detect the changes by its observation and replace the outdated frame with the new observation. In comparison, the 3D reconstruction-based methods need to use more complex algorithms to update the 3D representation incrementally, which would be computationally costly.
>
> We also acknowledge that how this architecture should be applied to a dynamic environment is an open and challenging question.
> ***
> ### Visual Hallucination in Navigation Tasks
>
> We would really like to share more of our insights on this question, but unfortunately, we have reached the limitation of response. In short, we have met with all sorts of visual hallucinations on image retrieval, grounding, and misalignment between images and texts. This is one of our key motivations to design the data synthesis pipeline, as in the abstract. This could be inferred from **Table 2 (line 230)**, where we compare the base model (InternVL2.5-8B) with the model after instruction fine-tuning (OWMM-VLM-8B). Besides, we will add more discussions and analysis in the appendix in the final version.
> ***
> We really thank the reviewer's dedication and profession in evaluation and we hope our response addresses your concerns. We are also looking forward to your responses and more discussions in the next phase.

---

> ### Author Response · Authors · 2025-08-04
> **Kind Reminder for Response**
>
> Dear Reviewer,
>
> We really thank you for your thoughtful evaluation and constructive suggestions. We also understand that you might be busy on other submissions in this discussion session.
>
> We would like to hear from you about our response. Please kindly let us know if there is anything still unclear or you want to further discuss certain points. We are happy to provide more information before the discussion portal closure.
>
> Sincerely,
>
> The authors

---

> ### Comment · Reviewer_v5R9 · 2025-08-06
>
> Thank you for your response.
>
> The following questions have been addressed:
>
> Q1: Here is my understanding—please correct me if I’m wrong. You randomly explore the environment and select frames randomly for pose graph construction. This component is not claimed as a contribution.
>
> Q3: Regarding dynamic and large-scale environments: as you noted, handling dynamic scenes and large environments is beyond the scope of this work.
>
> Q4: If I understand correctly, the use of frame sequences as input limits the scalability of the approach and makes real-world deployment more challenging. That said, I appreciate your discussion on future directions—it’s thoughtful and constructive.
>
> The following question remains partially unaddressed:
> Q2: VLN is a broad topic, and I find it hard to agree that your task is different. From my perspective, the difference lies more in specific task settings. For example, REVERIE requires grounding of target objects to be manipulated, whereas your task also outputs the action type. However, both do not directly interact with objects. In addition, pre-mapping is often considered part of the VLN setup [1,2]; the main difference seems to be how the pre-exploration information is used. Also, interactive VLN [3,4] is aligned with navigation plus high-level manipulation, similar to your setting. It would be helpful to position your work more clearly within the broader VLN literature in the related work section.
>
> Q5: On visual hallucination: This is mentioned as a key challenge in both the introduction and abstract, and is mentioned during in your response as a motivation for creating synthetic data. However, the connection feels weak. There is little elaboration on the specific types of hallucinations encountered or how these observations lead to particular design choices. Since “hallucination” is a broad concept, clarifying what exactly is observed and how it shapes the synthetic data design would significantly strengthen the argument.
>
>
> [1] Chen, K., Chen, J. K., Chuang, J., Vázquez, M., & Savarese, S. (2021). Topological planning with transformers for vision-and-language navigation. In Proceedings of the IEEE/CVF Conference on Computer Vision and Pattern Recognition (pp. 11276-11286).
>
> [2] Wang, Z., Li, M., Wu, M., Moens, M. F., & Tuytelaars, T. (2025). Instruction-guided path planning with 3D semantic maps for vision-language navigation. Neurocomputing, 625, 129457.
>
> [3] Shridhar, M., Thomason, J., Gordon, D., Bisk, Y., Han, W., Mottaghi, R., ... & Fox, D. (2020). Alfred: A benchmark for interpreting grounded instructions for everyday tasks. In Proceedings of the IEEE/CVF conference on computer vision and pattern recognition (pp. 10740-10749).
>
> [4] Min, S. Y., Chaplot, D. S., Ravikumar, P., Bisk, Y., & Salakhutdinov, R. (2021). Film: Following instructions in language with modular methods. arXiv preprint arXiv:2110.07342.
>
> Overall, I think the contributions of the paper are limited—particularly for a machine learning conference such as NeurIPS. And the discussion of related work insufficient. As such, I will maintain my initial score.

---

> > ### Author Response · Authors · 2025-08-08
> >
> > Dear Reviewer v5R9,
> > Thank you for your continued engagement and thoughtful follow-up comments. We apologize for the remaining unaddressed concerns due to the length limitation of the initial rebuttal box. Regarding the Q2 and Q5, we now have more space to provide a clearer explanations:
> >
> > ---
> > ## Q2: Positioning within VLN Literature
> >
> > We understand your concern about the positioning of our work within the broader VLN literature. We also agree that more thorough discussion of related work would strengthen our paper. Generally speaking, our work inherits most of the settings from OVMM/ OWMM, as discussed in Section 2 Related Works (line 79) and testing environment from HomeRobot[1] challenge in Neurips 2023 challenge. Homerobot also provides an extensive comprison for over 10 relevant environments/benchmarks, including Alfred as you mentioned. I would like to add this table to help explain the underlying differences:
> > | Environment          | Scenes  | Object Cats | Inst.     | Continuous Actions | Sim2Real | Robotics Stack | Open Licensing | Open Manipulation |
> > | -------------------- | ------- | ----------- | --------- | ------------------ | -------- | -------------- | -------------- | ----------------- |
> > | ALFRED           | 120     | 84          | 84        | ❌                  | ❌        | ❌              | ✅              | ✅                 |
> > | **OVMM + HomeRobot** | **200** | **150**     | **7,892** | ✅                  | ✅        | ✅              | ✅              | ✅
> >
> > Although we are not familiar with the ALFRED environment, HomeRobot team mentions quite some fundamental differences between the environments, including
> > 1) Agent Action Space. We are using the continuous action space with positional control interface. For example, in our experiments, we need to send robot joint positions to controller to control the robot state, as we discussed in Section 3.2 OWMM Agent (line 165) and Appendix F.1 Simulation (line 580). We convert the high-level actions which are seemingly similar to that of ALFRED to low-level control signals to control the real robot in the environment. For example, for the pick action, our agent needs to project the pick location in ego-centric RGB image to a 3D (x,y,z) location in space. Then the arm controller computes the joints action and actuate the robot. It seems that ALFRED uses a floating camera as the ego centric agent without further testing the viability of each action, but to only set the simulated world in the desired state. In this sense, we kindly disagree with the comment of "However, both do not directly interact with objects" from the reviewer.
> > 2) Sim2Real Gap. Considering the overly simplified settings in ALFRED and the implementation of robotics stack and continuous actions in HomeRobot, there is naturally a sim2real improvement for HomeRobot. From the author's intuition, one of the obvious gaps in the ALFRED-based works could be the viability of actions. For example, in ALFRED-based works, the agent of floating camera simply snaps the object without real checking, and in REVERIE, the model even only predicts the bounding box without interaction. However, in the real mobile manipulation task, the robot should properly navigate to the place so that object to pick falls into the arm workspace so that it can be picked up. We have addressed this problem by training the VLM to decide in what ego-centric observation, when robot can pick the object of interest and when the robot should precisely choose a navigation point to tune its spatial relationship with the object of interest. This can be seen in Appendix D.2 Analysis on the training data (specifically discussed in line 533). Because our data is generated and validated on HomeRobot which has a similar interface with real robot, and verification scheme with simulated robot control pipeline, the fine-tuned model can be deployed onto real robotic systems with minimum sim2real gap.
> >
> > However, we admit that we overlook the VLN related works as the reviewer you mentioned, and we would happily add these early works and how they differ from the latest OVMM/OWMM settings. We strongly agree that they are closely related to our work, that even HomeRobot has also made close comparisons but much recent mobile manipulation publications started to miss those early works. We will significantly expand our related work section to better position OWMM-Agent within this landscape, particularly emphasizing the difference between the aforementioned VLN works.
> >
> > [1] Yenamandra, Sriram, et al. "Towards open-world mobile manipulation in homes: Lessons from the neurips 2023 homerobot open vocabulary mobile manipulation challenge." Arxiv 2024
> >
> > (more follow-ups incoming)

---

> > ### Author Response · Authors · 2025-08-08
> > **Follow-up Comment on Q5**
> >
> > ## Q5: visual hallucination
> >
> > During our evaluation of the baseline models, we observed three notable deficiencies caused by hallucination and lack capability of VLMs:
> >
> > - Error location outputs for navigation and object grounding in embodied tasks. As shown in Table 2, InternVL2.5-8B achieves very low affordance success rates—0.05 (±0.19), 0.18 (±0.31), and 0.14 (±0.26)—even though our input prompts matched the question format used by InternVL2.5 during its visual instruction tuning on bounding-box output. This is caused by hallucination, like model's error detection on target object or receptacle, or mismatch on detection and bounding-box output.
> > - Hallucination on multi-image understanding. In Table 2, InternVL2.5-8B attains only 1.27% accuracy on image retrieval among eight candidate images. In image retrieval task, the VLM must locate the image that contains the target object or receptacle within a multi-image set. InternVL2.5-8B frequently hallucinates—running chain-of-thought (CoT) on an image where the target object is absent—and consequently outputs incorrect answers.
> >
> > - Insufficient decision-making for long-horizon embodied tasks. In Table 3, where we use GPT-4o as the task planner, the model frequently falls into a dead loop, due to limited ability to decompose long-horizon tasks and to interpret the robot’s historical context. For example, at time step t-1 GPT-4o issues a search scene frame command and records completion in the summarization. At step t, the target object is already visible from the head view and the correct action should be navigation. However, GPT-4o misreads the summarization and erroneously runs chain-of-thought(CoT),thinking it still does not see the target.Finally, GPT-4o outputs search scene frame again, causing a dead loop.
> >
> > To address these failure modes—stemming from model hallucinations and limited embodied capabilities—we designed a data-collection pipeline that fuses: (i) precise affordance bounding boxes for navigation, pick, and place; (ii) large-scale multi-image sets and correct chain-of-thought(CoT) of each set; and (iii) template-based reasoning and summarization prompts into a QA dataset, enabling models to "think" right according to its images and history inputs. Trained on this dataset, OWMM-VLM shows significant improvements in affordance precision for navigation/pick/place, multi-image understanding, and decision-making on embodied tasks.
> >
> > We value the comment from the reivewer as we also it is important to provide the why and how information about dataset synthesis. We plan to add this discussion to Appendix D. Details of Datasets in the final version.

---

> > ### Author Response · Authors · 2025-08-08
> > **Extra discussions on the contributions**
> >
> > We acknowledge your assessment regarding the limited contributions in machine learning. However, we believe our work makes several significant contributions appropriate for the application area of NeurIPS, which we select as our primary area.
> >
> > As in paper **line(69-75)**, our contributions and meaning for the field are:
> > - A unified framework for OWMM that integrates open language following, whole-scene understanding, ego-centric perception, planning, and control by modeling the task as a VLM multi-image multi-turn chatting problem, addressing fragmentation in the field. This is also acknowledged by Reviewer 8pbX, "The idea of unifying scene understanding, state tracking, and affordance prediction into a VLM agent is novel and well-motivated," as well as Reviewer hsT1, "The fine-tuned OWMM-VLM effectively integrates vision and language for embodied reasoning."
> > - A simulation-based agentic data synthesis pipeline for generating large-scale, physically consistent multimodal training data for embodied AI. By demonstrating VLM's strong sim-to-real transfer capabilities in decision-making and grounding, even when fine-tuned by purely sim-synthesized data, we enhance the value of simulated data synthesis for embodied AI. Reviewer 8pbx shares the same opinion: "Strong generalization to real-world environments, using only simulation data, demonstrates practical significance."
> > - Open-source SOTA foundational model as well as the training data and serve as useful tools for the community. This is recognized by reviewer QZc6, "The proposed dataset and the agent model are beneficial for OWMM-related research."
> >
> > After all, the HomeRobot challenge (NeurIPS 2023) as well as the previous one (Rearrangement challenge on NeurIPS 2022) both built on the Habitat simulator, which shares the close settings as ours, were welcomed by the machine learning conference of NeurIPS. In this regard, we believe the contributions of our paper are suitable for NeurIPS, especially the application area.
> >
> > Through this thread of discussions, we can feel the expertise of the reviewer, and we understand we miss quite a lot of discussions of contexts. We have tried our best to share contextual information and unclear details in implementation. We hope the reviewer can take this into consideration and re-evaluate our paper. We will revise as much as we can for the final revision. We thank the reviewer again for your time and active engagement in discussion.
> >
> > Kind Regards,
> >
> > The authors

---

> > ### Author Response · Authors · 2025-08-09
> >
> > Dear reviewer,
> >
> > We thank the expertise and efforts in the evaluation of our paper. As the deadline for authors to respond to comments approaches, we would like to hear your ideas about our latest responses and if your remaining concerns have been addressed.
> >
> > Kind regards,
> >
> > The authors

---

### Official Review · Reviewer_hsT1 · 2025-07-01

**Clarity:** 3
**Significance:** 3
**Originality:** 3
**Rating:** 5
**Confidence:** 4

**Summary:**

This paper proposed OWMM-Agent, which performs open-world mobile manipulation tasks. The core of the system is a VLM that is fine-tuned using a large-scale synthetic dataset generated in simulation. This model is capable of handling multiple sub-tasks by reasoning over multimodal inputs, including long-term scene memory and short-term robot state history. The finetuning enables state-tracking of the task, resulting higher task success rate. The proposed model shows strong performance in decision making and grounding, for both simulation and real-world environment.

**Questions:**

- The gap between object image retrieval (88.56%) and goal image retrieval (30.39%) is quite large. Could the authors elaborate on potential reasons for this discrepancy? For example, are goal receptacles more visually ambiguous, less salient, or more linguistically diverse?

**Ethical Concerns:**

["NO or VERY MINOR ethics concerns only"]

**Final Justification:**

I encourage the authors to revise the paper in line with the rebuttal and the reviewers’ comments.
This work presents a valuable contribution, but its presentation and framing could be improved.
I also share reviewer 8pbX’s concern regarding the “open-world” claim.

I believe these issues can be addressed through revisions for the final version.
Also, the rebuttal and the authors’ responses have resolved my concerns and questions.
Therefore, I maintain my original score.

**Limitations:**

yes

**Quality:**

3

**Strengths And Weaknesses:**

## Strengths

- The paper is well-written and presented the problem and methodology cleary.
- The fine-tuned OWMM-VLM effectively integrates vision and language for embodied reasoning.
- Extensive experiments are conducted, including comparisons with strong baselines, ablation studies, and analysis of training data effects.



## Weakness

- Failure analysis needed: While the proposed method shows promising results, the full task success rate is notably low, even for the best-performing model with huge number of parameters. I understand this is a difficult task, and since there are multiple subtasks so low success rate is kind of natural considering accumulation of errors. However, a detailed failure analysis would help the reader better understand the primary bottlenecks and guide future improvements.
- It would be beneficial to explicitly discuss why the proposed model avoids dead loops. I think the fine-tuned VLM model and the robot history helps avoid repetition, so it can naturally avoid dead loop while other can't. (Please correct me if I'm wrong). It would be nice to clarify about this with some examples.
- The overall system appears to operate in open-loop control. While the VLM can understand the current state and make the decision based on it, but it is unclear how the system handles failures.  What happens if the picking object fails? Is the fine-tuned VLM model able to recover itself?  A discussion or experiment showing whether the VLM can detect and recover from failure would be valuable.

---

> ### Author Rebuttal · Authors · 2025-07-31
>
> We sincerely thank the reviewer for the thoughtful evaluation and for recognizing the clarity of the writing, the embodied reasoning enabled by **OWMM‑VLM**, and the breadth of experiments. Below we address the requests on failure analysis, loop avoidance, and failure recovery, and then answer the question on the object–goal retrieval gap:
> ***
> ### Failure analysis in episodic evaluation
>
> We acknowledge that detailed failure analysis is crucial for understanding system limitations, and we plan to add more analysis discussions in the final version.  In fact, in **Table 3 in Section 5 (line 230)**, we have provided the cascaded success rate on the episodes, meaning until which step in the episode the agent is still successful, as detailed in **Section 5.2 (line 259)**.  And we have two key observations:
>
> 1) **Bottleneck exists on object picking/placing action.** While our model achieves strong performance in the early stages (88.56% on object image retrieval, 84.64% on robot navigation to object), performance degrades significantly in later stages (38.56% object picking, 30.39% goal image retrieval)
>
> 2) **Error accumulation matters.** Failed early actions compound, making later subtasks increasingly difficult.
>
> We will include this analysis in the revision and we plan to further dive into the first observation to check whether the object interaction action fails because of decision making (choosing wrong action) or grouding. We will keep you informed in the following discussion phase.
>
> We acknowledge that detailed failure analysis is crucial for understanding system limitations, and we plan to add more analysis discussions in the final version.  In fact, in **Table 3 in Section 5 (line 230)**, we have provided the cascaded success rate on the episodes, meaning until which step in the episode the agent is still successful, as detailed in **Section 5.2 (line 259)**.  And we have two key observations:
>
> 1) **Bottleneck exists on object picking/placing action.** While our model achieves strong performance in the early stages (88.56% on object image retrieval, 84.64% on robot navigation to object), performance degrades significantly in later stages (38.56% object picking, 30.39% goal image retrieval)
>
> 2) **Error accumulation matters.** Failed early actions compound, making later subtasks increasingly difficult.
>
> Besides, we further dive into episodic action sequences to analyze their failure reasons. Since the current evaluation pipeline does not support automatically failure case analysis, we randomly selected 100 failed samples from evaluation logs, and categorized their failures into four types:
>
> - **Ego-centric Decision Making Error**: Errors in pick/place or navigation actions, typically caused by the robot either being close enough but outputting a navigation action, leading it to move away and lose the target, or not being close enough but outputting a pick or place action.
> - **Image Retrieval Error**: The retrieved image ID is incorrect.
> - **Affordance Grounding (object/receptacle) Error**: The robot is close enough, but the center point of the output bounding box does not intersect with the target object/receptacle, causing a failed pick or place action, or the grounded object/receptacle is incorrect.
> - **Affordance Grounding (navigation) Error**: The output bounding box corresponds to an unreachable region or causes the target object/receptacle to disappear from the robot’s egocentric view.
>
> The evaluation results are as follows:
>
> | Model/Error ratio | Ego-centric Decision making Error（pick/place or navigation） | Image<br>Retrieval Error | Affordacne grounding(object/ receptacle) Error | Affordacne grounding(navigation) Error |
> | --- | --- | --- | --- | --- |
> | OWMM-VLM-38B | 36/100 | 28/100 | 15/100 | 21/100 |
>
>
> We will include these analyses and more visualizations in the revision appendix to provide more insight about the experimental results. We sincerely thank the reviewer for this constructive suggestion.
> ***
> ### Why our model avoids dead loops
>
> We really appreciate this insightful question and we mostly agree with the reviewer's interpretation of this phenomenon. We performed supervised fine-tuning (SFT) on a dataset that includes chain-of-thought (CoT) annotations. The CoT component enables the model to learn to understand tasks and historical information, make decisions by incorporating situational context, and summarize past information. With this capability improved, the model is better at solving sequential tasks: it can keep track of which stage it is currently at in completing the sequence, thereby avoiding repeatedly outputting the exact same action.
> ***
> ### Failure Recovery
>
> This is a good point that touches one of the **limitations of our current approach**. In fact, our system currently operates in open-loop mode where the VLM generates high-level actions based on observations but doesn't explicitly verify action success or implement recovery strategies. As discussed in the last point of deal loops, the current data synthesis pipeline focuses more on sequential reasoning and multi-modal action output, rather than failure recovery.
>
> However, we think to further strengthen failure recovery behavior and reduce error accumulation could be important **future work**. Recent progress on LLM Reinformance Learning [1] and observed "Aha moment"[2] reveal that this failure recovery behavior can be injected by an RL finetuning process: (i) explicitly synthesizing error‑restart training episodes in simulation; and (ii) exploring RL with language feedback for recovery policies.
>
> We would like to add this discussion to the Appendix B to discuss more about its limitations and possible future work.
> ***
> ### Gap between object image retrieval and goal image retrieval
>
> The figures the reviewer cites are drawn from different evaluation regimes. In our single‑step evaluation, the standalone ability of **object retrieval** and **goal (receptacle) retrieval** is comparable, the experiment results is presented at Table2.
>
> By contrast, the numbers in the episodic evaluation presented at Table3 compute success as “successful episodes / total episodes.” Under this protocol, **goal retrieval** success is conditioned on all preceding sub‑tasks, including a successful pick. For **OWMM‑VLM‑38B**, the episode‑level success for **goal image retrieval** is **30.39%**, and the preceding **object picked** success is **38.56%**. This confirms our point: the apparent gap at the episode level largely reflects upstream execution ceilings and error compounding, not an intrinsic disparity in standalone retrieval capability.
> ***
> We appreciate the reviewer’s constructive suggestions. We will incorporate the expanded failure analysis, the loop‑avoidance clarification with examples, the failure‑recovery discussion and ablation, and the clarified, conditioned retrieval metrics in the appendices and corresponding sections of the final version. We are also looking forward to your response or follow-up discussion in the next stage.
>
> [1] Guo, Daya, et al. "Deepseek-r1: Incentivizing reasoning capability in llms via reinforcement learning." Arxiv 2025.
>
> [2] Zhou, Hengguang, et al. "R1-Zero's" Aha Moment" in Visual Reasoning on a 2B Non-SFT Model." Arxiv 2025.

---

> ### Author Response · Authors · 2025-08-04
> **Kind Reminder for Response**
>
> Dear Reviewer,
>
> We really thank you for your thoughtful evaluation and constructive suggestions. We also understand that you might be busy on other submissions in this discussion session.
>
> We would like to hear from you about our response. Please kindly let us know if there is anything still unclear or you want to further discuss certain points. We are happy to provide more information before the discussion portal closure.
>
> Sincerely,
>
> The authors

---

> > ### Comment · Reviewer_hsT1 · 2025-08-05
> >
> > Thank you for the detailed responses.
> >
> > I'm consent with the rebuttal, and I would like to maintain my original score.
> > I encourage the authors to revise the final version according to the rebuttal.

---

> > > ### Author Response · Authors · 2025-08-06
> > >
> > > Dear Reviewer,
> > >
> > > We are glad that our responses have addressed your concerns! We would happily follow your suggestion and revise our final version.
> > >
> > > Thank you again for your thoughtful review and engagement, and we truly appreciate your time and support.
> > >
> > > All the best,
> > >
> > > Authors

---

### Official Review · Reviewer_QZc6 · 2025-07-03

**Clarity:** 1
**Significance:** 2
**Originality:** 2
**Rating:** 4
**Confidence:** 5

**Summary:**

This paper proposes a VLM-based agent for open-vocabulary mobile manipulation. The authors formulate the instruction-following problem as letting the VLM reason over history observations, scene graphs (pose graphs), and egocentric observations. To induce reasoning capabilities, the authors selected several proxy tasks for training the VLM-agent and generated the datasets correspondingly. The results show improvement on both real-world and synthetic environments.

**Questions:**

See the strength & weakness section. Currently it's missing some details and therefore I'm leaning towards a borderline reject decision (with the hope that the revision/rebuttal could significantly improve clarity on model design and dataset construction).

**Ethical Concerns:**

["NO or VERY MINOR ethics concerns only"]

**Final Justification:**

The rebuttal helps address major concerns on technical details, therefore I'm increasing my score to borderline accept.

**Limitations:**

yes

**Paper Formatting Concerns:**

No formatting concerns.

**Quality:**

2

**Strengths And Weaknesses:**

[+] The overall goal of this paper is well-motivated and is indeed an important problem to be studied.

[+] The proposed dataset and the agent model are beneficial for OWMM-related research.

[+] The authors tested not only in synthetic environments but also in real-world scenarios, indicating solid system design for real-world robots.

[-] One major concern about this paper is the potentially missing details about the whole pipeline. As the idea of using graphs and VLMs for reasoning in open-vocabulary mobile manipulation tasks has already been discussed before, the distinction between this paper lies in the details of the modules that improve the model performance (e.g. CoT design, pose graph construction for sim and real environments, etc.). However, both the main paper and the appendix lack sufficient discussion on these details, which hinders further understanding of the current good performance of the model. This also makes the current dataset construction and experimental analyses a bit plain without much insights in addition to performance improvements.

[-] I do feel the current version of paper writing could benefit from further revision and refinement.

---

> ### Author Rebuttal · Authors · 2025-07-30
>
> We really thank the reviewer for the thoughtful review of our paper. We appreciate your acknowledgment of the well-motivated research goal, the benefits of this paper for related research community, and the value of real-world experiments. Apart from the positive comments, the reviewer also raises some concerns about technical details about the module design. Although we would like to provide a revision for the questions mentioned to provide more details, this option is not allowed in the review program of this year. Instead, in the sections below, we will try our best to address these concerns.
> ***
> ## Chain-of-Thought (CoT) Design
>
> The CoT design is briefly demonstrated in Table 1 (line 183) of our paper, and we have uploaded examples in the supplementary materials. Besides, we have also provided an ablation experiment about the CoT process in **Appendix G (line 601)**, indicating that  performing reasoning during model generation and maintaining a summarization of historical information are very important for the success rate of the OWMM task; after removing them, the success rate drops significantly.But we also acknowledge the writing ambuity here and we plan to add a section to **Section 3.3** with the following details.
>
> Our CoT reasoning approach builds upon recent advances in embodied reasoning for robotics. Our OWMM-VLM generates structured reasoning chains that include: 1) Reasoning of the task instruction and summarization, and making decisions accordingly; 2) Performing perception, grounding, and task‑specific interpretation for the current egocentric-view image and the scene images. The model integrates visual information to support task decisions, for example, it determines whether the agent is close enough to an object in the current view, and then decides whether to perform pick/place or continue navigation; 3)Outputting the task decision and the execution targets/coordinates in the form of bounding boxes; 4)Summarizing the decision made at the current step and the actions executed; this summary is fed into the next step as input.
>
> The key insight of this design is that augmenting outputs with Chain‑of‑Thought (CoT), OWMM-VLM acquires patterns for task comprehension, scene perception, and decision‑making from structured training data. OWMM-VLM models also summarize historical context after each decision, allowing every subsequent step to jointly reason over prior history and current observations.
> ***
> ## Pose Graph Construction for Sim and Real Environments
>
> We really thank the reviewer for pointing out the missing details of pose graph contruction, as we also think thiese details are important for reproduction and real-world deployment. We aso We acknowledge the pose graph of images as multi-modal memory as one important agent design for global scene understanding, while meeting the VLM input data requirement as common VLM base models only accept texts and images as input. However, we do not claim pose graph construction method as one of our technical contributions, as in the Section 1 Introduction (line 68). As explained in **Section 3.1 line 136**, "... we assume a pre-mapping phase separating active exploration and the SLAM module from the OWMM task focus. This is practical, as most robotic vacuums automate room mapping before cleaning."
>
> We now provide more details about the implementation of pose graph construction. Basically, it involves pre-mapping stage and camera pose selection to cover the task-relavant images.
>
> **Pre-mapping Stage**: In simulated experiments, we get pre-mapping using the navigation mesh[1] provided by Habitat environment, where navigable locations are represented as mesh triangles. This pre-computed navmesh serves as a "map" for localization and navigation, similar to game engines. In the lab environment, we run RobiButler[2] over Fetch Lidar input for the pre-mapping stage and localize the robot geometrically. More specifically, RobiButler uses Gmapping[3] algorithm to compute the 2D occupancy map from lidar. The map is further used to localize the robot and camera poses.
>
> **Camera Pose Selection**: Considering the context length of current VLMs, we select 8 views from pre-mapping stage and 1 current ego-centric view in our experiments, as demonstrated in Figure 2. In simulation, as detailed in **Section 4.1 line 205**, we "first set the robot at the location of the receptacle where objects were initially located and the goal receptacle, sampling the robot's head-view images. Subsequently, we randomly positioned the robot and captured its head-view images." This ensures sufficient scene coverage while maintaining computational efficiency. In the real robot, we extract the keyframes from the ROS bag RGB data and manually select the frames related to the task to guarantee enough scene coverage.
>
> Besides, although actively selecting related frames from videos is beyond the contributions of this paper, there are some related works trying to address this issue. VideoAgent[4] and the follow-up works proposed to perform video understanding by retrieving task-relavant frames and then conducting a normal VLM reasoning process. Here we assume we have a ground truth VideoAgent and focus on downstream tasks.
>
> As in **Appendix B Limitations (line 455),** We acknowledge this is one of the limitations of our method and we plan to add more discussions to the final version of our paper to provide more insight. Addressing this limitation could be future work.
>
> [1] Arkin, Ronald C. "Path planning for a vision-based autonomous robot." *Mobile robots I*. Vol. 727. SPIE, 1987.
>
> [2] Xiao, Anxing, et al. "Robi Butler: Multimodal Remote Interaction with a Household Robot Assistant." ICRA 2025.
>
> [3] G. Grisetti, et al., “Improved techniques for grid mapping with rao-blackwellized particle filters,” IEEE transactions on Robotics, 2007
>
> [4] Wang, Xiaohan, et al. "Videoagent: Long-form video understanding with large language model as agent." ECCV 2024.
> ***
> ## Experimental Analysis and Dataset Construction
>
> For this section, we kindly disagree that "dataset construction and experimental analyses a bit plain without much insights in addition to performance improvements". In fact, we have provided rich details about dataset construction pipeline and the underlying insight. Besides, we have also conducted extensive ablation studies on both training data compositions and model designs.
>
> ### Dataset Construction Insights
>
> Our agentic data synthesis pipeline offers several key insights for the robotics community:
>
> Automated Template-Based Reasoning Framework: We demonstrate that high-quality robotic training data can be generated with minimal human annotation through the combination of following components: 1) Systematic PDDL-based task sequences and ground-truth symbolic world representations. This addresses a critical scalability challenge in robotics data collection, as detailed in **Section 4.1 (line 197)**. 2) Domain-Specific Re-labeling Strategy: The PDDL-based system can only generate symbolic state descriptions with poor diversity in predicates and label names based on the model dataset. We introduced a GPT-4o-based re-labeling process that generates diverse descriptions for object labels and states, suited for open language instruction. This insight shows how foundation models can be leveraged to enhance dataset diversity without manual effort, as described in **Section 4.2 (line 220)**.
>
> ### Experimental Analysis Insights
>
> Besides, our experimental analyses provide several non-trivial insights for 1) what is a good agentic fine-tuning dataset on our domain-specific task and 2) what are the effective components of our VLM model.
>
> In **Appendix** **D.2 Analysis on the training data (line 514)**, we studied scaling law in fine-tuning and dataset diversity. We systematically analyzed how object and scene diversity affect model performance, discovering that diversity has negligible effects on multi-modal capabilities (fluctuations within 5% range), while data volume shows logarithmic improvement patterns. This finding challenges conventional wisdom about dataset diversity requirements in robotics, and could be contributed to that most grounding capabilties derive from the base model and its large pre-training data. Besides, we also present two speculations from the observation in **Appendix D.2 (line 533)**: 1) Distance-based decision making prior can be learned in a data-driven way and 2) multi-image retrieval presents a bottleneck in our task. These insights could have broader implications for the field.
>
> In **Appendix G Ablation Study on OWMM-VLM (line 585),** we conducted the ablation study and found that 1) Grounding format matters. Our ablation study shows that bounding box prediction significantly outperforms direct coordinate prediction (0.93 vs 0.65 for object grounding), revealing that consistency with pre-trained VLM representations is crucial for effective fine-tuning. 2) CoT matters. We demonstrate that removing reasoning and summarization capabilities leads to substantial performance degradation across all metrics, providing concrete evidence for the importance of structured reasoning in embodied AI systems.
> ***
> ## Revision Plan
>
> We acknowledge that the current version could benefit from revision and will address this through:
> 1. Enhanced Technical Detail: We will expand Section 3 with more comprehensive implementation details for CoT design, pose graph construction and place them in the appendinx.
> 2. Improved Experimental Analysis: We will expand the analysis of our experiemnts to provide deeper insights into our design choices and their impact on performance.
> ***
> We are really grateful for the valuable suggestions from the reviewer and we hope our response addresses your concerns. We are also looking forward to your responses and more discussions in the next phase.

---

> ### Author Response · Authors · 2025-08-04
> **Kind Reminder for Response**
>
> Dear Reviewer,
>
> We really thank you for your thoughtful evaluation and constructive suggestions. We also understand that you might be busy on other submissions in this discussion session.
>
> We would like to hear from you about our response. Please kindly let us know if there is anything still unclear or you want to further discuss certain points. We are happy to provide more information before the discussion portal closure.
>
> Sincerely,
>
> The authors

---

> > ### Author Response · Authors · 2025-08-08
> > **Second kind reminder for response**
> >
> > Dear reviewer,
> >
> > We hope this message finds you well. This message kindly reminds you that the deadline for the authors to respond to comments is tomorrow, Aug. 8th, 11.59pm AoE. Please let us know your ideas about our rebuttal and if your questions have been addressed or not.
> >
> > We understand you are quite busy these days due to the heavy workload of rebuttal and discussions. We will also be happy to hear from you about further suggestions/discussions. The earlier you provide your comments, the better we can respond.
> >
> > Kind regards,
> >
> > The authors

---

### Official Review · Reviewer_8pbX · 2025-07-05

**Clarity:** 3
**Significance:** 4
**Originality:** 3
**Rating:** 5
**Confidence:** 2

**Summary:**

This paper proposes OWMM-Agent, a novel vision-language model (VLM)-driven agent for open-world mobile manipulation (OWMM) tasks. The core idea is to reformulate OWMM as a multi-turn, multi-image, and multi-modal reasoning problem. The proposed architecture comprises:
1. A unified VLM (OWMM-VLM) that performs global scene understanding, agent state tracking, and high-level action generation from multi-modal inputs;
2. An agentic data synthesis pipeline that generates scalable, instruction-conditioned training episodes in simulation, significantly reducing human annotation needs.
Experimental results show state-of-the-art performance across both simulation and real-world settings, demonstrating strong zero-shot generalization and outperforming GPT-4o-based baselines.

**Questions:**

1. Pre-Mapping Assumption: The system assumes a pre-built pose graph and scene images. Could the model be extended to perform online mapping and actuation in tandem?
2. Action Format: The model uses bounding boxes for affordance grounding. Are there failure cases where bounding boxes are insufficient ?

**Ethical Concerns:**

["NO or VERY MINOR ethics concerns only"]

**Final Justification:**

Pre-mapping Assumption: The authors provided a thoughtful and technically grounded response to the concern regarding the pre-mapping requirement. While the current system assumes a pre-mapped environment, the authors clarified that the system can be adapted to an online mapping setup with minimal changes. I find this clarification satisfactory and believe it resolves the primary concern.

"Open-World" Claim: The authors clarified that their use of the term “open world” refers to semantic diversity (e.g., unseen scenes, object categories, and instances), rather than unconstrained physical exploration. This aligns with common usage in prior work (e.g., HOMERobot challenge) and was transparently acknowledged in the paper’s limitations. I accept this framing, though a more prominent clarification in the main paper (rather than appendix) would have been helpful.

Action Format: The authors conducted an ablation study comparing bounding box vs. point-based action outputs, showing that box-based actions consistently perform better. Their explanation—pretraining alignment and robustness to noise—is reasonable. While rare failures exist, these are due to recognition issues rather than the representation itself. I consider this issue resolved.

The authors were responsive and precise in their rebuttal, providing additional implementation details and future plans. The raised concerns were addressed satisfactorily, and the paper’s core contributions remain novel. I maintain a positive score.

**Limitations:**

Yes.

**Paper Formatting Concerns:**

None observed.

**Quality:**

4

**Strengths And Weaknesses:**

Strengths:
1. The idea of unifying scene understanding, state tracking, and affordance prediction into a VLM agent is novel and well-motivated.
2. Strong generalization to real-world environments, using only simulation data, demonstrates practical significance.
3. Experimental results are extensive: they include both single-step evaluations (with clear affordance grounding metrics) and episodic evaluation, including real-world robot experiments.
Weaknesses:
1. The assumption of a pre-mapping phase (pose graph and RGB map) weakens the claim of “open world.”

---

> ### Author Rebuttal · Authors · 2025-07-31
>
> We sincerely thank the reviewer for the time and effort devoted to evaluating our paper and for recognizing the strengths of our work, in particular the unified VLM agent, simulated agentic data‑synthesis, and extensive experiments. Apart from positive feedback, the reviewer also raised some concerns about the pre‑mapping assumption and action format. We would like to address these concerns below:
> ***
> ## Response to Weaknesses
>
> ### Pre‑mapping and the “open‑world” claim
>
> We agree that assuming a pre‑mapping phase may appear to narrow the scope of “open world”, as we have also discussed in **Appendix B Limitations (line 455)**.  However,  in our formulation *open world* primarily refers to the semantic breadth of environments instances, with 200 scenes, 150 categories, and 7892 object instances, especially when the agent is tested under novel environments and objects, as detailed in **AppendixD.1 (line 477)**. We inherit this setting from previous works such as[1] from Meta, as discussed in **Section 2 Related Works (line 82)**.
>
> Besides, we think the current system can be adapted to online system without pre-mapping. For example, running on an online SLAM system, the posed‑image graph we maintain can also be updated online while the agent acts: because scene images are sampled from the egocentric stream, the agent can resample from the most recent frames during execution and replace images tied to already‑executed actions. To increase the diversity and utility of the maintained image set, we can re‑score candidate frames and refresh highly redundant views with more informative ones. *We will add this discussion—and an ablation comparing **static** vs. **online‑refreshed** image graphs—to the appendix of the final version.*
>
> [1] Yenamandra, Sriram, et al. "Towards open-world mobile manipulation in homes: Lessons from the neurips 2023 homerobot open vocabulary mobile manipulation challenge." Arxiv 2024.
> ***
> ## Answer to Questions
>
> ### Pre‑Mapping Assumption — Can the model be extended to online mapping?
>
> Yes. Our agent can be extended to **jointly update the posed‑image graph during execution** with minimal changes:
>
> - **Online refresh mechanism.** At each decision step, we resample from the newest egocentric frames captured during action execution and **swap out** frames corresponding to completed waypoints.
> - **Lightweight redundancy control.** We compute similarity scores between candidate frames and the current graph using efficient visual encoders (e.g., EfficientNet[2] or CLIP[3]) and **replace near‑duplicates** to preserve coverage and context without imposing heavy SLAM overhead.
> - **Practicality.** This keeps the memory/latency budget stable because we cap the graph size and trade redundant views for fresher, task‑relevant views.
>
> We will include implementation details (resampling policy, replacement thresholding) and a small‑scale study in the appendix, comparing online refresh vs. static pre‑mapping in terms of success rate and per‑step latency.
> ***
> ### Action Format — Are there failure cases where bounding boxes are insufficient?
>
> In **Appendix G**, we conduct an ablation on the action‑output representation (bounding box vs. point coordinates). Across OWMM tasks, **bounding boxes consistently yield higher success rates** than point‑coordinate outputs. We hypothesize two reasons: (i) the InternVL‑family pre‑training includes box‑grounded supervision, which better aligns with box outputs; and (ii) boxes carry local spatial context (extent/scale) that is more robust to calibration noise and detection jitter than a single point. **The detailed experiment results are in Table9 in Appendix G (line 604)**.
>
> **Failure cases.** We do observe *rare* failures with box outputs, typically when the target is an **uncommon or long‑tail object** (e.g., *Schleich Allosaurus*), where the error stems from **recognition/grounding** rather than the box representation itself.
> ***
> We appreciate the reviewer’s constructive feedback. We will incorporate the above clarifications in the final version. We also consider adapting the system to an online mapping setting could be meaningful future work. We are also looking forward to more discussions in the next phase.
>
> [2] Tan, M. & Le, Q. *EfficientNet: Rethinking Model Scaling for Convolutional Neural Networks.* ICML, 2019.
>
> [3] Radford, A. et al. *Learning Transferable Visual Models From Natural Language Supervision.* ICML, 2021.

---

> > ### Comment · Reviewer_8pbX · 2025-08-05
> > **Thanks for the rebuttal.**
> >
> > Pre-mapping Assumption: The authors provided a thoughtful and technically grounded response to the concern regarding the pre-mapping requirement. While the current system assumes a pre-mapped environment, the authors clarified that the system can be adapted to an online mapping setup with minimal changes. I find this clarification satisfactory and believe it resolves the primary concern.
> >
> > "Open-World" Claim: The authors clarified that their use of the term “open world” refers to semantic diversity (e.g., unseen scenes, object categories, and instances), rather than unconstrained physical exploration. This aligns with common usage in prior work (e.g., HOMERobot challenge) and was transparently acknowledged in the paper’s limitations. I accept this framing, though a more prominent clarification in the main paper (rather than appendix) would have been helpful.
> >
> > Action Format: The authors conducted an ablation study comparing bounding box vs. point-based action outputs, showing that box-based actions consistently perform better. Their explanation—pretraining alignment and robustness to noise—is reasonable. While rare failures exist, these are due to recognition issues rather than the representation itself. I consider this issue resolved.
> >
> > The authors were responsive and precise in their rebuttal, providing additional implementation details and future plans. The raised concerns were addressed satisfactorily, and the paper’s core contributions remain novel. I maintain a positive score.

---

> > > ### Author Response · Authors · 2025-08-06
> > >
> > > Dear Reviewer,
> > >
> > > We are glad that our responses have addressed your concerns! Thank you again for your thoughtful review and engagement, and we truly appreciate your time and support.
> > >
> > > All the best,
> > >
> > > Authors

---

> ### Author Response · Authors · 2025-08-04
> **Kind Reminder for Response**
>
> Dear Reviewer,
>
> We really thank you for your thoughtful evaluation and constructive suggestions. We also understand that you might be busy on other submissions in this discussion session.
>
> We would like to hear from you about our response. Please kindly let us know if there is anything still unclear or you want to further discuss certain points. We are happy to provide more information before the discussion portal closure.
>
> Sincerely,
>
> The authors

---

### Comment · Area_Chair_8toL · 2025-08-05
**Please Engage in the Discussion Period ASAP of the OWMM-Agent Paper**

Dear reviewers,

This is your AC. The authors have provided rebuttals to your reviews, and the discussion period is close to an end. Please respond to the authors whether your concerns have been addressed:

Reviewer 8pbX:
- Have your questions been clarified?

Reviewer QZc6:
- Do the new details provided about the pipeline address your concern?

Reviewer hsT1:
- Are you satisfied with the explanation about the performance gap?

Reviewer v5R9:
- Does the rebuttal change your mind?

Regards

-AC

---

### Comment · Area_Chair_8toL · 2025-08-05
**New Rules from the Program Chairs: Reviewers will need to post comments before acknowledgement!**

Dear reviewers,

Thank you for your initial reviews and now the program chairs have issued new rules to encourage participation in this period. Note that the author-reviewer discussion period has been extended by 48 hours.

Under the new rules, you will need to post at least a comment before submitting the “Mandatory Acknowledgement”. Otherwise, you will be **flagged for “Insufficient Review”**.

Here are some general guidelines for the author-reviewer discussion period:
+ It is not OK to stay quiet.
+ It is not OK to leave discussions till the last moment.
+ If authors have resolved your (rebuttal) questions, do tell them so.
+ If authors have not resolved your (rebuttal) questions, do tell them so too.

Thank you for your efforts!

Regards,

-Your AC

---

### Decision · Program_Chairs · 2025-09-17

**Decision:**

Accept (poster)

**Comment:**

## Summary

This paper introduces OWMM-Agent, a novel framework for open-world mobile manipulation. The core contributions are a unified vision-language model (OWMM-VLM) that integrates scene understanding, state tracking, and action generation, and a scalable, simulation-based agentic data synthesis pipeline to fine-tune the model. This approach aims to address the challenges of generalization and system complexity in mobile manipulation by reformulating the task as a multi-modal, multi-turn reasoning problem. The authors demonstrate state-of-the-art performance in both simulation and real-world zero-shot scenarios.

The paper received a mixed set of reviews, with three reviewers ultimately leaning towards acceptance and one recommending rejection.

## Strengths identified by the reviewers
- The unified framework that combines perception, reasoning, and action generation into a single VLM is novel and well-motivated.

- The agentic data synthesis pipeline is a significant contribution, enabling scalable data generation and demonstrating strong sim-to-real transfer.

- The experimental evaluation is extensive, covering both simulation and real-world tests with strong baseline comparisons.

## Weaknesses
- Reviewer 8pbX (Accept) and Reviewer hsT1 (Accept) raised concerns about the pre-mapping assumption, the "open-world" claim, and the need for more detailed failure analysis. The authors provided thorough rebuttals, including additional ablation studies and clarifications on their methodology and claims. Both reviewers were fully satisfied with the responses, confirmed their concerns were resolved, and maintained their positive scores.

- Reviewer QZc6 (Borderline Accept) initially found the paper lacking in technical details regarding the implementation pipeline (e.g., CoT design, pose graph construction). The authors' rebuttal supplied these missing details, which addressed the reviewer's concerns and led them to raise their score from borderline reject to borderline accept.

- Reviewer v5R9 (Borderline Reject) raised more fundamental concerns about the paper's novelty and positioning within the existing Vision-Language Navigation (VLN) literature. Despite a detailed, multi-part rebuttal and follow-up discussion from the authors, the reviewer remained unconvinced. They argued that the task setting was a simplified variant of existing VLN tasks and that the connection between the stated problem of "visual hallucination" and the proposed data synthesis solution was not sufficiently justified. The reviewer maintained their negative score, deferring the final decision to the Area Chair.

## Recommendation Justification
This paper tackles the challenging and important problem of open-world mobile manipulation. The consensus among the majority of reviewers is that the proposed unified agent architecture and the agentic data synthesis pipeline are novel and significant contributions. The authors' ability to achieve strong zero-shot performance on a real robot using only synthetic data is a notable result with high practical impact for the robotics and embodied AI communities.

The primary dissenting opinion from Reviewer v5R9 centers on the paper's positioning relative to prior VLN work. While this is a valid point and the paper would benefit from a more detailed discussion of this context, I believe the work's focus on integrating both navigation and manipulation with continuous control and its strong sim-to-real results distinguish it sufficiently. The contributions in creating a capable, open-source foundational model and dataset for mobile manipulation are valuable for the field.

The paper is technically solid, the experiments are thorough, and the contributions have the potential for high impact. While the concerns of Reviewer v5R9 are noted, the overwhelming positive feedback and the successful resolution of most issues lead me to believe the paper is a clear accept.